



# Historical CO₂ emissions from land-use and land-cover change and their uncertainty

Thomas Gasser[1], Léa Crepin[1,2], Yann Quilcaille[1], Richard A. Houghton[3], Philippe Ciais[4], Michael Obersteiner[1,5]

[1] International Institute for Applied Systems Analysis (IIASA), 2361 Laxenburg, Austria
[2] AgroParisTech, 75231 Paris, France
[3] Woods Hole Research Center, 02540 Falmouth, Massachusetts, USA
[4] Laboratoire des Sciences du Climat et de l'Environnement, LSCE/IPSL, Université Paris-Saclay, CEA – CNRS – UVSQ, 91191 Gif-sur-Yvette, France
[5] Environmental Change Institute, University of Oxford, OX1 3QY Oxford, UK

Correspondence to: Thomas Gasser (gasser@iiasa.ac.at)

**Abstract.** Emissions from land-use and land-cover change are a key component of the global carbon cycle. Models are required to disentangle these emissions and the land carbon sink, however, because only the sum of both can be physically observed. Their assessment within the yearly community-wide effort known as the Global Carbon Budget remains a major difficulty, because it combines two lines of evidence that are inherently inconsistent: bookkeeping models and dynamic global vegetation models. Here, we propose a unifying approach relying on a bookkeeping model that embeds processes and parameters calibrated on dynamic global vegetation models, and the use of an empirical constraint. We estimate global CO2 emissions from land-use and land-cover change were $1.36\pm0.42$ PgC yr-1 (1-σ range) on average over 2009–2018, and $206\pm57$ PgC cumulated over 1750–2018. We also estimate that land-cover change induced a global loss of additional sink capacity – that is, a foregone carbon removal, not part of the emissions – of $0.68\pm0.57$ PgC yr-1 and $32\pm23$ PgC over the same periods, respectively. Additionally, we provide a breakdown of our results' uncertainty following aspects that include the land-use and land-cover change data sets used as input, and the model's biogeochemical parameters. We find the biogeochemical uncertainty dominates our global and regional estimates, with the exception of tropical regions in which the input data dominates. Our analysis further identifies key sources of uncertainty, and suggests ways to strengthen the robustness of future Global Carbon Budgets.





## 1. Introduction

The annual flux of carbon dioxide to the atmosphere caused by land use and land cover change (LULCC) is a key part of the global carbon budget (Friedlingstein et al., 2019) (GCB). It is one of the two historical anthropogenic sources of $CO_2$ 
(along with fossil fuel burning and industry emissions), and when added to the land carbon sink it gives the net land-to-atmosphere carbon exchange. In fact, it is so closely connected to the land carbon sink that choosing incompatible definitions for these two fluxes can lead to double counting or missing part of the budget (Gasser and Ciais, 2013). Models are required to disentangle these emissions and the land carbon sink, however, because only the sum of both can be physically observed. The GCB2019 assessment (Friedlingstein et al., 2019) estimated LULCC emissions were $1.5\pm0.7$ PgC $yr^{-1}$ (1-σ range) on 
average over 2009–2018. This value relied on two lines of evidence: dynamic global vegetation models (DGVMs), that are complex process-based and spatially explicit models of the terrestrial carbon cycle (and related processes), and bookkeeping models, that are parametric models that convolute time-series of LULCC areal perturbations with empirical response functions describing changes in ecosystem carbon stocks after these perturbations.

The strengths and weaknesses of those two types of models are opposite: the DGVMs are developed to precisely describe 
the biogeochemistry of plants and ecosystems albeit without overly focusing on LULCC, whereas bookkeeping models are specifically designed to evaluate LULCC emissions but without any explicit representation of biogeochemical processes. Any comparison between those models is rendered even more difficult by two factors. First, DGVMs and bookkeeping models do not naturally follow the same definition of LULCC emissions, as DGVMs tend to include the "loss of additional sink capacity" (LASC) in their estimate (Gasser and Ciais, 2013; Pongratz et al., 2014). The LASC is defined as the 
difference between the actual land sink under changing land cover and the counterfactual (stronger) land sink under preindustrial land cover. The LASC, however, is not an actual physical flux: it is a foregone carbon removal. (A few DGVMs are now capable of providing LULCC emissions that are consistent with the bookkeeping definition, however, but these estimates are not used to establish the GCB's best-guess estimates (Friedlingstein et al., 2019).) The second source of discrepancy is the different historical LULCC data sets used to drive the models. In the GCB2019, the DGVMs and one of 
the two bookkeeping models (Hansis et al., 2015) used spatially explicit LULCC drivers from the land use harmonization (LUH) project (Hurtt et al., 2011; Hurtt et al., 2006). The second bookkeeping model (Houghton and Nassikas, 2017), however, used independent driving data compiled from national statistics of the UN's Food and Agriculture Organization (FAO), and especially from its 2015 Forest Resources Assessment (FAO, 2015) (FRA2015).

Here, using the OSCAR reduced-form Earth system model, we bridge the gap between these approaches and estimates. 
OSCAR embeds a bookkeeping module as well as simplified biogeochemical processes calibrated on DGVMs, which makes it a valuable tool to consistently bridge across the different estimates used in the GCB, as illustrated in Table 1. The goal of this paper is threefold. First, it is to provide another bookkeeping estimate of global and regional LULCC emissions – hopefully to be used in the future GCB – obtained with an original model. Second, it is to revise and investigate further the LASC estimates we provided in an earlier version of the GCB (Le Quéré et al., 2018b). Third, it is to investigate the





uncertainty range in both these fluxes along the three axes of analysis shown in Table 1: inclusion of the LASC, driving LULCC data sets, and biogeochemical parameterization.

## 2. Overview of the methodology

OSCAR is built to emulate the behavior of more complex (typically three-dimensional and process-based) models. Its land carbon cycle is calibrated on DGVMs and spatially aggregated over broad world regions (Gasser et al., 2017). A brief

description of its land carbon cycle is provided in Appendix A1. The preindustrial steady-state is based on simulations made for the GCB (also called the TRENDY exercise), and the transient responses of net primary productivity, wildfires and heterotrophic respiration to changes in atmospheric $CO_2$ and climate are on CMIP5 simulations (Arora et al., 2013). Its bookkeeping module keeps track of ecosystems affected by LULCC separately, offering a consistent and easy way to isolate LULCC emissions from the land sink (Gasser and Ciais, 2013; Gasser et al., 2017). LULCC activities accounted for are

gross land-cover change transitions, wood harvest (without land-cover change), and shifting cultivation (i.e. rapid rotations between young natural ecosystems and cropland). OSCAR does not include fire as a land management tool (Houghton et al., 2012), emissions caused by the draining and burning of peatlands (Carlson et al., 2015; Guillaume et al., 2018; Houghton and Nassikas, 2017), or the impact of LULCC on the export of terrestrial organic carbon to the ocean through the land-ocean aquatic continuum (Regnier et al., 2013). Here, we use OSCAR v3.1: an iteration over v3.0 in which the land carbon cycle's

structure was slightly altered, and its preindustrial steady-state recalibrated. Both changes are described in Appendix A2, and older changes that led from v2.2 to v3.0 are summed up in Appendix A3. OSCAR v2.2 was comprehensively described by (Gasser et al., 2017). Earlier versions have been used previously to investigate LULCC emissions (Arneth et al., 2017; Bastos et al., 2016; Eglin et al., 2010; Gasser and Ciais, 2013; Gitz and Ciais, 2003).

We follow an experimental protocol similar to the one used in the recent GCBs (and fully described in Appendix A4). The

model is driven with observed changes in environmental conditions (global atmospheric $CO_2$, regional temperature and precipitation), and with specific LULCC driving data. Thanks to the model's flexibility and low computing requirements, we also run different LULCC data sets, sensitivity experiments in which either changes in environmental conditions or LULCC are turned off, and a Monte Carlo ensemble of 10,000 different biogeochemical parameterizations. Our best-guess estimate is derived by combining results obtained with two LULCC data sets: the latest iteration of the LUH2 data set used for the

GCB2019 (Friedlingstein et al., 2019), and the FRA2015 data set used by (Houghton and Nassikas, 2017). The latter ends in 2015, however, so we extended its results with constant values over 2016–2018 equal to the average of the 2011–2015 period. To constrain this best-guess ensemble, each of the 20,000 elements is given a weight based on how well it compares to a reference value of the net change in land carbon stock between 1850 and 2018 (see Appendix A5). All values presented in this study are the resulting weighted averages and weighted standard deviations. The constraining value is calculated as

the cumulative fossil-fuel emissions minus the change in atmospheric and oceanic carbon stocks over the period – all taken from the GCB2019.



## 3. Results

### 3.1. Global LULCC emissions and LASC

Our primary results are shown in Table 2 and Figure 1. We find global LULCC emissions of 1.36±0.42 PgC yr⁻¹ on average over 2009–2018, which is consistent with the GCB2019 estimate (Friedlingstein et al., 2019) of 1.5±0.7 PgC yr⁻¹. Our reported value follows a bookkeeping definition (Gasser and Ciais, 2013; Pongratz et al., 2014) and is therefore comparable to that of the GCB. We simulate that historical LULCC emissions peaked in 1959, at a value of 1.61±0.55 PgC yr⁻¹. Since then, they have remained roughly steady, but reached a local minimum in 1999 of 1.14±0.52 PgC yr⁻¹. Overall, we estimate that a total of 206±57 PgC was emitted between 1750 and 2018, and 178±50 PgC between 1850 and 2018. These values are also consistent with the GCB2019 estimates of 235±75 PgC and 205±60 PgC over the same periods, respectively.

We estimate a global loss of additional sink capacity of 0.68±0.57 PgC yr⁻¹ on average over 2009–2018. It amounted to a cumulative 32±23 PgC between 1750 and 2018, and 31±22 PgC between 1850 and 2018. This extremely low difference is explained by the nature of the LASC. It is a foregone land carbon sink; a product of both land-cover change and environmental condition changes. Since, environmental conditions were only marginally changing during the early 1750–1850 period, the land sink and the LASC were extremely low. As this change in environmental conditions became more intense in the recent past, both fluxes also increased in intensity. Our new estimates of the LASC are larger than those we reported in a past GCB (Le Quéré et al., 2018b), owing to the change in empirical constraint. Table 2 shows that we can obtain estimates similar to the older ones by reverting back to the old constraint (that is, the land sink without LULCC simulated by DGVMs). Performance of this alternative constraint is further presented in Figure A1.

The effect of the constraint is further detailed in Figure 1c. Constraining corrected the overestimate and substantially reduced the range of the cumulative net land-to-atmosphere flux over 1850–2018: from an unconstrained value of −4±84 PgC to −22±29 PgC (compared to the constraining value of −25±30 PgC). Applying the constraint essentially resulted in the exclusion of aberrant values of the land carbon sink without significantly affecting LULCC emissions. The cumulative LULCC emissions over 1850–2018 were indeed 176±48 PgC before constraining (and 178±50 PgC after). The stronger constraining effect on the land sink is logically visible on the LASC, however, as the unconstrained cumulative LASC over 1850–2018 was 25±23 PgC (and the constrained one is 32±23 PgC).

### 3.2. Comparison to GCB models

Comparability between our best guess estimates and those of the GCB2019 is limited (because of differing definitions or driving data), and we therefore dedicate this section to comparing like to like. Figure 2a compares the annual land sink over 1959–2018 in the absence of LULCC perturbation (i.e. with a preindustrial land cover). OSCAR simulates a slightly larger land sink by the end of the period than DGVMs, although it remains within their uncertainty range. It also reproduces fairly well the inter-annual variability of the complex models. Note that this specific simulation is used by the GCB to define the land sink. This implies that their land sink is not comparable to ours (except in this figure), since theirs does not include the





LASC. Figure 2b compares LULCC emissions calculated using the DGVMs' definition (Gasser and Ciais, 2013; Pongratz et

al., 2014) (i.e. including the LASC) over 1959–2018. OSCAR is in line with the DGVMs, although it estimates a slightly larger flux over the beginning of the period. More importantly, it displays a much lower uncertainty range than the spread among DGVMs. Since OSCAR emulates well the carbon densities of the DGVMs (Appendix, and Table A1), we attribute this difference in spread to the large variance in the land cover map used by the DGVMs (Table A2) and their processing of the input LULCC data set.

Figure 2c compares OSCAR and BLUE estimates of the bookkeeping LULCC emissions (i.e. without the LASC). BLUE is one of the two bookkeeping models used in the GCB2019, and both models are driven by the LUH2-GCB2019 data set. OSCAR and BLUE display similar annual variations in their LULCC emissions, but BLUE is systematically higher than OSCAR, and above the 1-σ range of our estimates by the end of the simulation. Figure 2d compares OSCAR and H&N estimates (also without the LASC). H&N is the second bookkeeping models of the GCB2019, and both models are this time

driven by the FRA2015 data set. Again, both models display similar annual variations, except near the end of the simulation, and this time H&N is systematically lower than OSCAR, although it remains mostly within its 1-σ range. Given that BLUE and H&N are parameterized with the same carbon densities, one would expect that OSCAR's estimates would systematically be either higher or lower. The fact that it is not the case suggests part of the differences between the three bookkeeping models comes from other factors, possibly such as structural assumptions or ways of processing and implementing the

LULCC data sets.

### 3.3. Uncertainty analysis

Although we cannot investigate the aforementioned structural differences between bookkeeping models, our experimental setup allows investigating several factors within OSCAR that affect the spread in our global results. Figure 3 and Table 3 summarize this. The first factor is whether the LASC is included in the estimate of LULCC emissions, as it is usually the

case when they are calculated with DGVMs, which is illustrated in Figure 3a. The difference between including and excluding the LASC corresponds, over the last decade, to a debiased 1-σ range of ±0.43 PgC yr$^{-1}$ and a coefficient of variation (CV) of ±25% (see Appendix A6). This rather substantial value is in line with previous studies that quantified this discrepancy (Gasser and Ciais, 2013; Stocker and Joos, 2015). Because the LASC became non-negligible only in the recent past, the effect of its inclusion or exclusion on cumulative LULCC emissions is smaller than for recent annual emissions: we

estimate it is only ±20 PgC (±9%) over 1750–2018. It is however crucial to understand that the intensity of this discrepancy will keep increasing and accumulating as long as changes in environmental conditions do not stabilize (Gasser and Ciais, 2013). This is illustrated in Figure 3a by the positive trend in the CV. In our view, this ever-growing discrepancy strongly pleads in favor of choosing, retaining and consistently applying one clear definition of the LULCC emissions. In the following, we exclude the LASC from LULCC emissions, and therefore discuss it separately.

The second factor of uncertainty is the driving LULCC data set. Figure 3b shows the difference between the average bookkeeping emissions estimates based on LUH2-GCB2019 and those based on FRA2015. We find the annual emissions



from the two data sets are in particularly good agreement on average over the last decade (Table 2), although this is purely fortuitous as the discrepancy is ±0.30 PgC yr$^{-1}$ (±24%) over 1995–2004, and even peaks at ±0.39 PgC yr$^{-1}$ (±34%) in 1999. More worrying, perhaps, is the two data sets' disagreement on the trend in emissions after 1990. This discrepancy is hidden
in our best-guess emissions that are rather even over the last 30 years. In terms of cumulative emissions over 1750–2018, however, results from the two data sets are in good agreement, with only a ±8 PgC (±4%) discrepancy. Additionally, Figure 3c and 3d display the same source of uncertainty but among different versions of each of the two main data sets. This variation caused by updating the data sets is visible, for instance, when comparing older versions of the GCB together. We find that the difference among several versions of the same data set is of the same order of magnitude as the one between our
two main data sets. For the LUH data set, this is explained by several factors, from the simple update of the historical land-cover data used as input (Klein Goldewijk et al., 2017; Klein Goldewijk et al., 2011) to the complete overhaul of how shifting cultivation is estimated (Heinimann et al., 2017). Among the FRA-based data set's versions, this difference is found to be somewhat larger. This likely owes to the concomitant update of some biogeochemical parameters of the H&N model (Houghton and Nassikas, 2017) that we cannot separate here, because the results shown in Figure 3d are not based on
OSCAR.

The third and last factor of uncertainty is the parameterization of the model (for biogeochemistry). Through our Monte Carlo ensemble, we find a weighted standard deviation of ±0.40 PgC yr$^{-1}$ (±29%) for annual emissions averaged over 2009–2018, and of ±55 PgC (±27%) for emissions cumulated over 1750–2018. Except in some specific years, this source of uncertainty in annual emissions is the largest of the three we studied, and it dominates without exception in cumulative
emissions. Carbon densities (and parameters determining them) are the key modeling factors explaining this spread (Gasser and Ciais, 2013). Figure 3f and Table 2 show the spread in our results when looking only at the variation caused by the parameters that relate to harvest wood products (HWPs). It is found to be one order of magnitude smaller than the total uncertainty caused by all parameters, confirming that biogeochemical parameters explain most of the uncertainty. However, we acknowledge that OSCAR likely underestimates the HWP-related uncertainty, because there is only one option to choose
from, in the Monte Carlo setup, as to how HWPs are split between pools with different decay timescales (Appendix A7).

Finally, a similar uncertainty breakdown for the LASC is reported in Table 2. We find that between our two main LULCC data sets, the uncertainty in the average annual LASC over the last decade is ±0.21 PgC yr$^{-1}$ (±31%); and it is ±10 PgC (±31%) for the cumulative LASC since year 1750. The much higher CV in cumulative LASC compared to that in cumulative LULCC emissions suggests that the latter is kept relatively low thanks to compensation effects that do not come into play in
the former. We also find that the biogeochemical uncertainty in the LASC is high: ±0.50 PgC yr$^{-1}$ (±77%) for the annual flux over 2009–2018, and ±19 PgC (±63%) for the cumulative flux over 1750–2018. Those values reflect the large uncertainty in the ecosystems' response to transiently changing environmental conditions, despite our constraining (unconstrained CVs are ±98% and ±86%, respectively).





### 3.4. Breakdown by region

Figure 4 and Table 4 provide our best-guess estimates of the bookkeeping LULCC emissions in our ten broad world regions. Without surprise, tropical regions (Latin America, Sub-Saharan Africa, and South and Southeast Asia, in decreasing order) are found to be the main LULCC emitters over the last decade, with a positive trend over the last 50 years. Conversely, North America, Europe, Former Soviet Union, and China are all found to have a decreasing trend over the last 50 years, to the point of North America, Europe and China being net carbon absorbers over the last decade. Looking at a

larger historical period, Latin America and South and Southeast Asia were the top two emitters over 1750–2018, with North America being the third one. It must be noted, however, that because of uncertainties this ranking is not statistically significant. When the subset of our simulations driven by the FRA2015 data set is isolated, our estimates compare very well with that of H&N (Houghton and Nassikas, 2017), (Table A3).

The uncertainty in our regional bookkeeping LULCC emissions can be attributed to the LULCC data sets and the

biogeochemical parameters using Figure 4 and Table 4. For North America, Former Soviet Union, and to a lesser extent Europe, the two LULCC data sets lead to emissions that are in rather good agreement, which implies the regional uncertainty is dominated by the biogeochemical parameterization. In tropical regions, however, the two data sets show substantial disagreement, to the point of being the main source of uncertainty in Sub-Saharan Africa, and in South and Southeast Asia. Remarkably, the disagreement in the emissions of Latin America around year 1990 seems to explain the global opposite

trends at this date shown in Figure 3b. China is another region in which the discrepancy between the two data sets leads to a substantial uncertainty range. On the one hand, FRA2015 exhibits large-scale forest plantation in China based on national declarations, which leads to a significant atmospheric carbon removal. On the other hand, the LUH2-GCB2019 ignores those declarations and considers that – over the same period – China lost a large amount of forest. Ultimately, it is not the goal of this paper to provide a detailed analysis of the regional discrepancies between the two data sets, or to recommend one over

the other. Nevertheless, we produced Figure A2 showing regional LULCC drivers, to offer a starting point for such an endeavor.

Our estimate of the LASC is also broken down regionally in Figure 4. The annual LASC of most regions follows a similar trend as the global one. In North Africa and Middle East, and Oceania, however, the noise produced by the inter-annual variability appears to dominate over the trend. It is unclear what exactly causes this noise, but the fact that both regions

include large non-vegetated areas suggests that the parameterization of OSCAR is not very robust in such a case. The noise intensity in Oceania is even larger than the signal in any other region, suggesting that some of the uncertainty in our LASC estimates could be an artefact of this weakness in our modeling approach. As to the regional split of the cumulative LASC over 1750–2018, it follows roughly that of the cumulative LULCC emissions, although it is modulated by the land sink's relative efficiency in each region. Latin America is the region in which the largest part of this loss of sink capacity occurred

(almost one third), followed by North America, South and Southeast Asia, and Sub-Saharan Africa. Uncertainties in the LASC are too high, however, for this ranking to be determined with good statistical confidence.



### 3.5. Breakdown by transition

Figure 5 and Table 5 show a breakdown of our global bookkeeping LULCC emissions and LASC following seven categories of LULCC activities. Forest-related land-cover change dominates historical bookkeeping emissions. Over the last decade, we estimate an average of 1.86±0.57 PgC yr⁻¹ was emitted by deforestation for establishing cropland, an additional 0.55±0.26 PgC yr⁻¹ was from other types of deforestation (e.g. for pastoral land, or simply forest degradation), and a capture of −1.36±0.49 PgC yr⁻¹ came from reforestation and afforestation. These estimates include the effect of our shifting cultivation driver that encompasses traditional activities such as "slash-and-burn", which leads to large but compensating gross carbon fluxes caused by back-and-forth deforestation/reforestation activities (Houghton et al., 2012; Li et al., 2018; Yue et al., 2018). The cumulative emission over 1750–2018 was 213±93 PgC by deforestation for cropland, 77±27 PgC by other types of deforestation, and 60±33 PgC through the loss of other natural land. This was partly compensated by −144±99 PgC from reforestation and afforestation, and −20±18 PgC when other natural land was regained.

The uncertainty in the bookkeeping LULCC emissions is largely dominated by the discrepancy between the two main LULCC data sets. For annual emissions, this is even reinforced by the fact that shifting cultivation is included in our estimates. Figure A3 indeed shows that both data sets have a very different level of shifting cultivation area, although this is somewhat artificial as it is caused by the difference in the data sets' starting year. Therefore, our uncertainty ranges for deforestation and reforestation categories are overestimated. For cumulative emissions, however, this overestimation is much lower, since in OSCAR shifting cultivation has a long-term effect of zero net emissions (Appendix A7). Notwithstanding this, a few other clear differences between the two data sets remain, such as the opposite trend in 1990 in the "other deforestation" category, or the difference in wood harvest.

When we split the LASC between these transitions types, we obtain a slightly different picture. Our three categories of natural land appropriation caused roughly similar amounts of cumulative LASC: 14±6 PgC from deforestation for cropland, 15±6 PgC from other deforestation, and 24±25 PgC from loss of other natural land. This was partially compensated by negative LASC (i.e. increase in sink capacity) of −7±5 PgC caused by reforestation and afforestation, and −3±4 PgC caused by other natural land gain. Other types of LULCC led to negligible LASC. As to the uncertainty in the LASC, the noise we identified in the previous section can be attributed to the "other natural land" biome. Combined to the diagnosis from the previous section, this suggests OSCAR could maybe benefit from separating desert areas (i.e. bare soils) from the Non-Forest biome. However, this would make processing the LULCC data sets more difficult, as new assumptions should then be made as to how much of this new biome is affected by LULCC.

### 3.6. Breakdown by carbon pool

A final axis of analysis OSCAR can provide is a breakdown along the model's carbon pools, and therefore indirectly following its biogeochemical processes. Figure 6 shows such a breakdown into our three main carbon pools: vegetation carbon, soil carbon, and HWPs. Bookkeeping LULCC emissions over the last decade consisted in a combination of





−3.66±0.96 PgC yr$^{-1}$ of vegetation carbon (i.e. biomass) regrowth, 2.84±0.85 PgC yr$^{-1}$ emitted by equilibrating soils, and
2.18±0.65 PgC yr$^{-1}$ emitted by HWPs being oxidised. The equilibration of soils here includes both the heterotrophic
respiration in originally carbon-rich soils that is not compensated by enough primary productivity (e.g. when deforesting to
establish cropland) and the oxidation of slash products (i.e. of dead biomass left on site after land-cover change). The three
sub-fluxes have been steadily increasing over the past century or so. Cumulated over 1750–2018, these three pool-specific
values amount to −443±155 PgC, 373±137 PgC, and 276±84 PgC, respectively.

For the LASC, this pool-based decomposition concerns only the vegetation and soil pools, as no HWPs are involved in the
processes driving the land sink. Both components of the annual LASC are positive, with notable inter-annual variability, and
with positive trend, reaching 0.33±0.25 PgC yr$^{-1}$ for the vegetation and 0.35±0.36 PgC yr$^{-1}$ for soils, on average over 2009–
2018. Over 1750–2018, the cumulative component fluxes are 16±7 PgC for the vegetation and 16±16 PgC for soils. These
positive values must be interpreted as a storage of carbon that did not happen, because the preindustrial land-cover was
modified and the new ecosystems could not provide as strong a land sink as the preindustrial ones. The breakdown shows
how this storage would have been split between vegetation and soil carbon pools, had it occurred. Since it is implicitly
determined by the model's processes, this breakdown is heavily model-dependent, and largely dominated by the
biogeochemical uncertainty.

## 4.   Discussion

OSCAR satisfactorily emulates the carbon densities and stocks of DGVMs (Table A1), but these stocks are in the lower
end of existing assessments. The DGVMs we calibrated OSCAR upon have global preindustrial pools of 457±77 PgC for the
vegetation and 1140±336 PgC for the soil, whereas the IPCC fifth assessment (Ciais et al., 2013) reports 450–650 PgC and
1500–2400 PgC, respectively. Some of the difference in soil carbon may be explained by the existence of "passive" soil
carbon that is not mobilized under the timescales we consider here (Barré et al., 2010), and that may not have been reported
or modeled by the DGVMs. Nevertheless, the relatively low carbon pools suggest that our bookkeeping LULCC emissions
could be underestimated. Alternatively, our preindustrial land-cover taken from the LULCC data sets could be inaccurate
(Table A2). We ran the LUH2 data set starting in 850, and did not find substantial carbon loss between 850 and 1750 (23±15
PgC in total). Other studies specifically focused on the older past have found much higher carbon loss over that early period
(Erb et al., 2018; Kaplan et al., 2011; Pongratz et al., 2008), again suggesting this part of our results could be
underestimated.

A key feature of OSCAR is that the model's carbon densities transiently change as a response to changes in environmental
conditions. This change in carbon densities is fully coupled to the bookkeeping module, and therefore impacts bookkeeping
LULCC emissions. This feature contrasts with the fixed carbon densities of other bookkeeping models, and it makes possible
accounting for processes such as $CO_2$-fertilisation, wildfire changes, and climate feedbacks. Without any change in
environmental conditions, we find annual bookkeeping LULCC emissions would have been 1.11±0.35 PgC yr$^{-1}$ on average





over the last decade, and cumulative emissions would have been 191±52 PgC over 1750–2018. This is respectively 19% and 7% less than our best guesses with environmental changes. The effect on cumulative emissions is in line with an older estimate (Gasser and Ciais, 2013). The effect on annual emissions, however, is higher. This suggests that this effect increases with time, and will keep increasing in the future, as environmental conditions change and get further away from the

preindustrial ones. This underscores the importance of building hybrid bookkeeping models like OSCAR capable of capturing such an effect.

A structural limitation of this version of OSCAR is the absence of any age-specific process. This means that none of the model's parameters depends on the time elapsed since a given LULCC perturbation (i.e., it has no age classes). For instance, 5-years old forests grow and die at the same rate as 50-years old ones. This, by construction, means that the biomass

regrowth of disturbed ecosystems follows an exponential response curve, which we acknowledge is unrealistic. The impact of this structural choice is difficult to estimate, but since it affects only dynamics and not carbon densities, we can speculate annual emissions are more impacted than cumulative emissions. Other regrowth curves could be introduced (Fekedulegn et al., 1999), although it would require introducing age-dependent functions in the model's formulation, which would in turn make it heavier and slower. Actually, when OSCAR v2.4 was developed, the only process that had been age-dependent until

then, namely the decay of HWPs (Gasser et al., 2017), was reformulated to be age-independent. The reason for this simplification is that, beyond being a carbon-cycle model, OSCAR is also an Earth system model, and the complexity of each of its modules has to be kept in check. It is not excluded, however, that future variants of the model will see implementation of such a feature.

A final structural element that we find worth mentioning is the biome aggregation of our model. The final list of five

biomes in OSCAR (Appendix A2) is a trade-off between the PFTs of the DGVMs and the land-cover classes of the LULCC data sets. Typically, DGVMs tend to focus on natural ecosystems (i.e. they have many types of forests), while LULCC data sets focus on anthropogenic ecosystems (i.e. more types of croplands and pastures). Our list of biomes aimed at limiting the number of assumptions made when processing both types of data for implementation within OSCAR, but some were necessary nonetheless. Qualitatively, we see two important caveats caused by our biome aggregation. First, since we have

only one natural biome to cover all natural land but forests, we average actual natural ecosystems with relatively high carbon densities such as shrubland with almost carbon-empty ecosystems like deserts. We saw in previous sections that this may explain part of the large uncertainty in our LASC estimates, but it likely also affects our bookkeeping LULCC emissions. Second, we do not distinguish between primary and secondary natural land. In other words, pristine and disturbed natural ecosystems are assumed to have the same steady-state carbon densities. This does not mean that the actual carbon densities

are the same, however. It means that it is assumed that, if left undisturbed, previously disturbed ecosystems will grow back to the exact same steady state as that of never-disturbed ones. Because they relate to carbon densities, these structural aspects are likely to have the largest impact on our estimates. Quantifying it would require a significant amount of work, however, and it would undoubtedly require making new assumptions that would in turn introduce additional uncertainty and potential biases.





**5.  Conclusion**

In spite of those caveats, this study has introduced an innovative method to estimate historical LULCC emissions and LASC, whereby a bookkeeping approach, data from processed-based models, several LULCC data sets and an empirical constraint are consistently combined. We have also identified key sources of uncertainty that must be reduced to improve future GCBs. One easy improvement is to decide on where to account for the LASC. We argued elsewhere (Gasser and

Ciais, 2013) that it is ill-advised to include the LASC into LULCC emissions, because it is a theoretical flux that cannot be observed and that does not tend to zero after LULCC activities cease. Reducing the other sources of uncertainty is a more challenging endeavor, however. Although satellite data (Hansen et al., 2013) and crowd-sourcing (Fritz et al., 2019) are a promising way to establish more accurate land-cover maps in present days, these must be backcast over the past to be relevant for the long-term dynamic of the global carbon cycle. Such backcasting generates new uncertainty (Peng et al.,

2017), and additional data perhaps in the form of historical records (Bastos et al., 2017; Houghton and Nassikas, 2017) is required to mitigate the lack of direct observations. We found the biogeochemical uncertainty dominated, although that is a reflection of the uncertainty in the DGVMs' own parameterization, and not the one stemming from direct observations of real-life carbon densities. Improving the DGVMs' calibration, for instance by assimilating observational data, is an obvious albeit resource-intensive way of reducing this source of uncertainty. Posterior evaluation and weighting of the DGVMs is

another approach, be it through a specifically designed protocol such as ILAMB (Collier et al., 2018) or a synthesis setup like ours.





## A. Appendices

### A.1. Brief description of the land carbon cycle module

The land carbon cycle module of OSCAR v3.1 is used in "offline" mode: it is not coupled to the rest of the Earth system, and in particular, permafrost carbon release (Gasser et al., 2018) is not accounted for. The global terrestrial biosphere is divided in pairs of regions and biomes, noted (*i,b*), representing the "average" biome *b* of the *i*-th region with assumed homogeneous biogeochemical characteristics. The module is therefore not spatially explicit, and the regional aggregation for this study follows the ten regions defined in (Houghton and Nassikas, 2017) (their Table 2). A detailed analytical description

of the module is provided  hereafter in Appendix A7.

The first part of the module describes the evolution of vegetation, litter and soil carbon densities (i.e. carbon stocks per unit area) and the areal carbon exchanges between these pools and/or the atmosphere, within each set (*i,b*) and in the absence of LULCC. The preindustrial steady-state values of these carbon densities and areal fluxes are calibrated on DGVMs. During a transient simulation, these values are affected by environmental conditions: changes in atmospheric $CO_2$ concentration

impact net primary productivity (NPP) – the "fertilization" effect – and wildfire intensity, while changes in regional yearly temperature and precipitation alter NPP, wildfire intensity and heterotrophic respiration rate.

The second part of the module describes the effect of LULCC using a bookkeeping approach. When a LULCC perturbation occurs, carbon from the originally undisturbed (*i,b*) pools is redistributed to other pools, including an anthropogenic pool of harvested wood products (HWPs). The new pools can also be within another biome *b'* in the case of

land-cover change. This displaced carbon follows the biogeochemical properties of the new pools, thus slowly tending toward a new steady-state. Following the discussion and recommendation of (Gasser and Ciais, 2013), the carbon fluxes and pools of these transitioning ecosystems are defined as a difference to their expected but yet-to-be-reached new steady-state, so that the effect of any LULCC perturbation tends toward zero on the long run. This corresponds to "definition 3" of (Gasser and Ciais, 2013) and to "definition B" of (Pongratz et al., 2014).

### A.2. Recalibration of the preindustrial land carbon cycle


The carbon cycle in each combination of region and biome (*i,b*) is represented by a three-box model, illustrated in Figure A3. In OSCAR v3.1, the three-box model was slightly altered but remains very close to that of earlier versions (Gasser et al., 2017). Concretely, a flux going directly from the vegetation carbon pool to the soil carbon pool (and therefore bypassing the litter carbon pool) was added ("$f_{mort2}$" in Figure A3). This simple change enables using the three-box model as a two-box

model without changing its structure or equations. In turn, this enables emulating complex models that do not provide enough information to be properly emulated with the three-box model without making any additional assumption. In addition to this increased flexibility, the model was extended with two new fluxes: emissions from harvested crop products ("$e_{harv}$"), and emissions caused by pasture grazing ("$e_{graz}$").





In OSCAR v3.1, the parameters describing the preindustrial steady-state of the land carbon cycle module were recalibrated on outputs of the DGVMs that took part in the GCB2018 (Le Quéré et al., 2018a) (i.e. the TRENDYv7 models). We used outputs from the control experiment (named "S0" in their protocol), in which no LULCC occurs and environmental conditions such as atmospheric $CO_2$ and climate are maintained at their preindustrial level, to calibrate the parameters of the natural biomes ("Forest" and "Non-Forest"). However, because it follows preindustrial (here, 1700) land-cover data, the area extent of anthropogenic biomes ("Cropland", "Pasture" and "Urban") is very low in the S0 experiment. For these anthropogenic biomes, in order to avoid any bias in their parameters potentially caused by this low land-cover fraction, we decided to use the last years of another experiment instead (namely, "S4"), in which historical LULCC occurs but environmental conditions are still maintained at preindustrial levels. We acknowledge the existence of LULCC in S4 is not in line with our aim of calibrating a steady-state, but it is a necessary compromise to have a large enough area extent of anthropogenic biomes for proper calibration. Note that some DGVMs did not provide enough data and could therefore not be used for calibration at all, which led to a total of 11 models used out of the 16 original DGVMs (see Table 1). Detailed calibration protocol is given in Appendix A8.

Table A1 shows a comparison of the global net primary productivity and carbon pools in the OSCAR v3.1 and in the original DGVMs. The emulation is not perfect, and the preindustrial global (and a fortiori regional) carbon pools of OSCAR do not exactly match those of the emulated DGVMs, for three main reasons. First, since we average and homogenize biogeochemical properties over large world regions, we lose some accuracy, as unevenly distributed carbon pools are not explicitly represented in OSCAR. This bias is somewhat reduced by defining regions that show a certain bio-climatic consistency (e.g. separated tropical regions). Second, OSCAR works with clearly defined biomes, and we therefore have to map the DGVMs' plant functional types (PFTs) onto the biomes of our model. Since few DGVMs provide detailed fluxes and pools on a PFT basis, we further have to use an ad hoc method to distribute aggregated variables between our biomes (see detailed calibration protocol). Third, we calibrate carbon densities and not stocks, and some discrepancy is introduced by the fact that we do not use the DGVMs' preindustrial land-cover map (Table A2). Despite these three caveats, Table A1 demonstrates that the emulation remains largely satisfactory.

### A.3. Changes between OSCAR v2.2 and v3.1

**v3.1.** Changes to the land carbon cycle are described hereinabove (Appendix A2). The new structure made it necessary to adapt the wetlands module, so that $CH_4$ emissions from wetlands now scale with the relative change in total heterotrophic respiration (and not the change in litter respiration).

**v3.0.1.** An error in the overlap function for the radiative forcing of $CH_4$ and $N_2O$ was corrected. This error appeared during the rewriting of v3.0 and did not affect earlier versions.

**v3.0.** OSCARv3 was completely recoded from scratch in Python 3 (instead of Python 2), with an entirely new structure and solving scheme. This version heavily relies on the "xarray" Python library (Hoyer and Hamman, 2017) to parallelize Monte Carlo simulations and/or scenarios. The default solving scheme was changed to a forward-Eulerian exponential integrator.




All underpinning physical equations and parameter values remain the same as in v2.4. Both versions were compared, and no significant difference was found beyond the effect of the new solving scheme.

**v2.4.** This version was developed as a bridge version between v2 and v3. Our goal was that v2.4 be as close to v3.0 as
possible, without changing the overall structure of the model, at this point. The preindustrial land-cover map was changed to being that of the LULCC data set used to drive the model. The dependency of the fractional area of wetlands to the preindustrial land-cover map was removed, taken as the average of all previous parameterizations. The possibility of having HWPs follow a non-exponential decay was removed. To compensate, beyond the original parameterization of the HWP lifetimes (called "normal"), two new options were added in which these lifetimes are rescaled by 0.5 ("fast") or by 1.25
("slow"). Finally, the biome aggregation was fixed to that of v3: the five biomes described in this paper.

**v2.3.1.** A number of minor errors were fixed, and additional adjustments were made. Land carbon-cycle parameters for the urban biome were corrected. So were one parameterization for the radiative efficiency of tropospheric $O_3$, and one for the semi-direct effect of BC aerosols. One parameterization for the effect of $N_2O$ on stratospheric $O_3$, and one for fractional release factor of ozone depleting substances were removed. One parameterization for the radiative efficiency of tropospheric
$O_3$ was added. All these changes had almost no impact on the model's performance. Two additional changes had more impact, however. First, the discretization of the response functions for HWPs was corrected, which led to higher LULCC emissions after correction. Second, a new parameter was introduced to account for the fact that too much of the HWPs were assumed burnt. This amount was reduced by half, leading to better endogenous non-$CO_2$ biomass burning emissions, but having no impact on the $CO_2$ budget.

**v2.3.** The permafrost carbon model described by (Gasser et al., 2018) was implemented in the model's main branch.

**v2.2.2.** A minor error in one parameterization for the lifetimes of POA and BC was fixed. This had very little impact on the overall performance of the model.

**v2.2.1.** Two errors in the code were fixed. The first was in the function linking the surface ocean carbon pool to the surface ocean partial pressure in $CO_2$, which was leading to a too efficient ocean carbon sink under high warming and high
atmospheric $CO_2$. The effect was however negligible under historical conditions. The second was in the function linking atmospheric CO2 and surface ocean pH change. This function was not described by (Gasser et al., 2017); it was taken from (Bernie et al., 2010) but was incorrectly implemented.

**v2.2.** This version was comprehensively described by (Gasser et al., 2017).

### A.4.  Experimental setup

The land carbon cycle module of OSCAR is driven by: (i) global atmospheric $CO_2$ concentrations over 1700–2018 created for the GCB2019 exercise (Friedlingstein et al., 2019); (ii) observation-based reconstructions of regional air temperature and precipitation over 1901–2018 from the CRU-TS v4.03 (Harris et al., 2014) (iii) several LULCC data sets detailed hereafter. Atmospheric $CO_2$ before 1700 is assumed to be constant and equal to the preindustrial value used in OSCAR (Gasser et al., 2017), which is a very slight deviation from the GCB protocol, as our model's preindustrial reference year is 1750 and not





1700. Climate variables are offset by their average over the 1901–1930 period, assuming this corresponds to a preindustrial climate that is further extended backward before 1901. This is similar to the GCB protocol with the exception that we use the average of 1901–1930, whereas the GCB recycles the individual years of this period (leading to a 30-year cycle).

Our best-guess estimates are based on two significantly differing LULCC data sets. The first one is an update of the LUH2 data set made for the GCB2019 exercise, in which only the years past 1950 differ slightly from the original data set. The

second one is the data set used and created by (Houghton and Nassikas, 2017) on the basis of FAO and FRA2015 data. Although these two data sets have several data sources in common, they remain mostly independent given the way they internally process these input data. Additionally, for Figure 3, we ran simulations with older variants of the LUH data set. These are: the first land use harmonization (LUH1) data set (Hurtt et al., 2011) originally produced for the CMIP5 modeling exercise; an updated version of LUH1 made for the GCB2015 exercise, that was based on then-preliminary HYDE3.2 data

(Klein Goldewijk et al., 2017) instead of HYDE3.1 data (Klein Goldewijk et al., 2011); and the original LUH2 data set produced for CMIP6, as well as its two "Low" and "High" variants (Lawrence et al., 2016). All the data sets required some slight processing described in Appendix A9. Our simulations start in the earliest year of the LULCC data sets: in 850 for LUH2 variants, in 1500 for LUH1 variants, and in 1700 for FRA2015. This significantly differs from the GCB protocol that starts in 1700.

**A.5. Constrained Monte Carlo ensemble**

All those simulations were made following a probabilistic Monte Carlo setup in which 10,000 sets of the model's parameters are drawn randomly (with equiprobability). The combination of Monte Carlo elements, LULCC data sets and variant runs led to a total of 140,000 simulations and about 100 million simulated years. We constrained this large ensemble to limit the bias and spread that typically results from using OSCAR in such a probabilistic fashion. This is done in a way

similar to what we did for the GCB2017 (Le Quéré et al., 2018b). Each element of the Monte Carlo ensemble is given a weight ($w$) equal to:

$$w(x) = \frac{1}{\sigma\sqrt{2\pi}} \exp\left(-\frac{(x-\mu)^2}{2\sigma^2}\right) \tag{1}$$

where $\mu$ and $\sigma$ are the mean and standard deviation of the constraint, respectively, and $x$ is the value of the corresponding variable for this element of the ensemble. The constraint we use in this study is the cumulative net land-to-atmosphere flux

over 1850–2018, calculated as the residual of the carbon emitted by fossil-fuel burning and industry minus the carbon stored in the atmosphere and the ocean. These values were taken from the GCB2019 (Friedlingstein et al., 2019), their Table 8).

**A.6. Debiased uncertainty**

To analyze and separate each uncertainty factor, we average our simulations ensemble over all uncertainty axes but the one investigated, and then quantify the absolute (standard deviation) and relative (coefficient of variation; CV) uncertainty along





the remaining axis. Because the size of the remaining ensemble can be as small as only two elements, both values are debiased by multiplying them by a factor $\kappa_\sigma$ (Brugger, 1969):

$$\kappa_\sigma = \sqrt{\tfrac{N-1}{2}} \, \Gamma\left(\tfrac{N-1}{2}\right) / \Gamma\left(\tfrac{N}{2}\right) \tag{2}$$

where $N$ is the size of the remaining ensemble, and $\Gamma$ is the gamma function. This approach differs from a proper variance decomposition, but it is simpler to handle given the number of simulations we performed, and it does provide a relative ranking of the importance of each uncertainty factor.

### A.7. Analytical description of the land carbon cycle module

Following Figure A3, the evolution of vegetation ($c_{veg}$), litter ($c_{soil1}$) and soil ($c_{soil2}$) carbon densities is determined by a number areal fluxes: net primary productivity (npp), emission from wildfire ($e_{fire}$), emission from harvested crop products ($e_{harv}$), emissions from grazing ($e_{graz}$), mortality to litter ($f_{mort1}$) and to soil ($f_{mort2}$), metabolization from litter to soil ($f_{met}$), and heterotrophic respiration from litter (rh$_1$) and soil (rh$_2$). Using superscripts $i$ and $b$ to note regions and biomes, respectively, and a dot on top of a variable to note its first time differential, the associated differential system for all ($i,b$) is:

$$\dot{c}_{veg}^{i,b} = npp^{i,b} - e_{fire}^{i,b} - e_{harv}^{i,b} - e_{graz}^{i,b} - f_{mort1}^{i,b} - f_{mort2}^{i,b} \tag{3}$$

$$\dot{c}_{soil1}^{i,b} = f_{mort1}^{i,b} - f_{met}^{i,b} - rh_1^{i,b} \tag{4}$$

$$\dot{c}_{soil2}^{i,b} = f_{mort2}^{i,b} + f_{met}^{i,b} - rh_2^{i,b} \tag{5}$$

Each of these fluxes is then formulated as follows:

$$npp^{i,b} = \eta^{i,b} \, \mathcal{F}_{npp}^{i,b}(CO_2, T^i, P^i) \tag{6}$$

$$e_{fire}^{i,b} = \iota^{i,b} \, \mathcal{F}_{fire}^{i,b}(CO_2, T^i, P^i) \, c_{veg}^{i,b} \tag{7}$$

$$e_{harv}^{i,b} = \epsilon_{harv}^{i,b} \, c_{veg}^{i,b} \tag{8}$$

$$e_{graz}^{i,b} = \epsilon_{graz}^{i,b} \, c_{veg}^{i,b} \tag{9}$$

$$f_{mort1}^{i,b} = \mu_1^{i,b} \, c_{veg}^{i,b} \tag{10}$$

$$f_{mort2}^{i,b} = \mu_2^{i,b} \, c_{veg}^{i,b} \tag{11}$$

$$f_{met}^{i,b} = \mu_{met}^{i,b} \, \mathcal{F}_{rh}^{i,b}(T^i, P^i) \, c_{soil1}^{i,b} \tag{12}$$

$$rh_1^{i,b} = \rho_1^{i,b} \, \mathcal{F}_{rh}^{i,b}(T^i, P^i) \, c_{soil1}^{i,b} \tag{13}$$

$$rh_2^{i,b} = \rho_2^{i,b} \, \mathcal{F}_{rh}^{i,b}(T^i, P^i) \, c_{soil2}^{i,b} \tag{14}$$

where the three functions noted with script F are sensitivity functions to atmospheric $CO_2$, regional air temperature ($T^i$) and precipitation ($P^i$), all calibrated on CMIP5 models (Arora et al., 2013) and described in earnest by (Gasser et al., 2017).

The Greek letters introduced in Equations (6–14) are the parameters of the system: $\eta$ is the preindustrial NPP; $\iota$ is the preindustrial wildfire intensity; $\varepsilon_{harv}$ and $\varepsilon_{graz}$ are the export fractions from crop harvesting and animal grazing respectively; $\mu_1$ and $\mu_2$ are the mortality rates to litter and soil, respectively; $\mu_{met}$ is the metabolization rate; $\rho_1$ and $\rho_2$ are the preindustrial





heterotrophic respiration rates of litter and soil, respectively. Mathematically, these nine parameters are sufficient to define
the preindustrial steady-state of the system, noted with subscript 0:

$$c_{veg,0}^{i,b} = \eta^{i,b}/(\iota^{i,b} + \epsilon_{harv}^{i,b} + \epsilon_{graz}^{i,b} + \mu_1^{i,b} + \mu_2^{i,b}) \tag{15}$$

$$c_{soil1,0}^{i,b} = (\mu_1^{i,b} c_{veg,0}^{i,b})/(\rho_1^{i,b} + \mu_{met}^{i,b}) \tag{16}$$

$$c_{soil2,0}^{i,b} = (\mu_2^{i,b} c_{veg,0}^{i,b} + \mu_{met}^{i,b} c_{soil1,0}^{i,b})/\rho_2^{i,b} \tag{17}$$

These nine parameters are the ones recalibrated on the GCB2018 models.

In OSCAR, the LULCC perturbation is represented by three anthropogenic forcings: land-cover change ($\delta A_{cover}$), wood
harvest ($\delta H_{wood}$), and shifting cultivation ($\delta A_{shift}$). The first forcing describes area transitions from one biome to another, and
it is therefore defined along two biome axes $b$ and $b'$ representing the initial and final biomes of the transition. It is the only
forcing that actually alters the area extent ($A_{land}$) of the different biomes, following:

$$\dot{A}_{land}^{i,b} = \sum_{b'} \delta A_{cover}^{i,b' \to b} - \sum_{b'} \delta A_{cover}^{i,b \to b'} \tag{18}$$

The second forcing describes biomass harvested from woody biomes that then regrow, and it is defined along only one
biome axis. The third forcing describes reciprocal area transitions between one natural biome and another anthropogenic
biome, typical of practices such as slash-and-burn. It is defined along two axes, but its matrix representation in the ($b,b'$)
space is symmetrical, which implies a net zero land-cover change. To account for this last forcing in a computationally

efficient way, one key simplification is made in OSCAR. Shifting cultivation is assumed equivalent to harvesting the
biomass of ecosystems that are $\tau_{shift}$ years old (Gasser et al., 2017). This amount of harvested biomass is calculated as the
vegetation carbon density multiplied by the $\delta A_{shift}$ driver and by a reduction factor ($p_{shift}$) based on Eq. (13):

$$p_{shift}^{i,b} = 1 - \exp(-(\eta^{i,b}/c_{veg,0}^{i,b}) \tau_{shift}) \tag{19}$$

To manage the bookkeeping itself, OSCAR keeps track of LULCC-perturbed extensive variables as a difference to the

steady-state they would reach after a long enough time period (and that is described in the previous section). These variables
use uppercase letters (in opposition to the lowercase letters of the previous section), and have the subscript "$bk$" to denote
they are under bookkeeping. It is also necessary to introduce a new carbon pool for harvested wood products ($C_{hwp}$) that is
itself split into several subpools noted with superscript $w$. The differential system describing this part of the model is:

$$\dot{C}_{bk,veg}^{i,b} = NPP_{bk}^{i,b} - E_{bk,fire}^{i,b} - E_{bk,harv}^{i,b} - E_{bk,graz}^{i,b} - F_{bk,mort1}^{i,b} - F_{bk,mort2}^{i,b} + \delta C_{bk,veg}^{i,b} \tag{20}$$

$$\dot{C}_{bk,soil1}^{i,b} = F_{bk,mort1}^{i,b} - F_{bk,met}^{i,b} - Rh_{bk,1}^{i,b} + \delta C_{bk,soil1}^{i,b} + F_{slash1} \tag{21}$$

$$\dot{C}_{bk,soil2}^{i,b} = F_{bk,mort2}^{i,b} + F_{bk,met}^{i,b} - Rh_{bk,2}^{i,b} + \delta C_{bk,soil2}^{i,b} + F_{slash2} \tag{22}$$

$$\dot{C}_{hwp}^{i,b,w} = F_{hwp}^{i,b,w} - E_{hwp}^{i,b,w} \tag{23}$$

Equations (20–23) introduce new fluxes that correspond to the initialization step of the bookkeeping. $\delta C_{bk,veg}$, $\delta C_{bk,soil1}$ and
$\delta C_{bk,soil2}$ represent the initial carbon in the vegetation, litter and soil pools, respectively, as a difference to their respective

future steady-state. For the vegetation pool, it is assumed that the new ecosystems start without any biomass:

$$\delta C_{bk,veg}^{i,b} = -\sum_{b'} c_{veg}^{i,b} \delta A_{cover}^{i,b' \to b} - \delta H_{wood}^{i,b} - \sum_{b'} c_{veg}^{i,b} p_{shift}^{i,b} \delta A_{shift}^{i,b' \to b} \tag{24}$$





For the litter and soil pools, it is assumed that they start with the carbon of the old ecosystems:

$$\delta C_{bk,soil1}^{i,b} = \sum_{b'} c_{soil1}^{i,b'} \, \delta A_{cover}^{i,b'\to b} - \sum_{b'} c_{soil1}^{i,b} \, \delta A_{cover}^{i,b'\to b} \tag{25}$$

$$\delta C_{bk,soil2}^{i,b} = \sum_{b'} c_{soil2}^{i,b'} \, \delta A_{cover}^{i,b'\to b} - \sum_{b'} c_{soil2}^{i,b} \, \delta A_{cover}^{i,b'\to b} \tag{26}$$

From the old ecosystems, the above-ground biomass fraction ($\pi_{agb}$) is partly harvested and allocated to harvest wood product pools ($F_{hwp}$), following pool-specific allocation coefficients ($\pi_{hwp}$):

$$F_{hwp}^{i,b} = \sum_{b'} \pi_{hwp}^{i,b',w} \, \pi_{agb}^{i,b'} \, c_{veg}^{i,b'} \, \delta A_{cover}^{i,b'\to b} + \pi_{hwp}^{i,b,w} \, \delta H_{wood}^{i,b} + \sum_{b'} \pi_{hwp}^{i,b',w} \, \pi_{agb}^{i,b'} \, c_{veg}^{i,b'} \, p_{shift}^{i,b'} \, \delta A_{shift}^{i,b'\to b} \tag{27}$$

The fraction of biomass of the old ecosystem that is left on site ($p_{slash}$) is made of the rest of the above-ground biomass and the below-ground biomass:

$$p_{slash}^{i,b} = \left(1 - \pi_{agb}^{i,b}\right) + \left(1 - \sum_w \pi_{hwp}^{i,b,w}\right) \tag{28}$$

This defines the so-called "slash" fluxes to the litter ($F_{slash1}$) and soil ($F_{slash2}$) pools:

$$F_{slash1}^{i,b} = \sum_{b'} \frac{\mu_1^{i,b'}}{\mu_1^{i,b'}+\mu_2^{i,b'}} p_{slash}^{i,b'} \, c_{veg}^{i,b'} \, \delta A_{cover}^{i,b'\to b} + \frac{\mu_1^{i,b}}{\mu_1^{i,b}+\mu_2^{i,b}} p_{slash}^{i,b} \, \delta H_{wood}^{i,b} + \sum_{b'} \frac{\mu_1^{i,b'}}{\mu_1^{i,b'}+\mu_2^{i,b'}} p_{slash}^{i,b'} \, c_{veg}^{i,b'} \, p_{shift}^{i,b'} \, \delta A_{shift}^{i,b'\to b} \tag{29}$$

$$F_{slash2}^{i,b} = \sum_{b'} \frac{\mu_2^{i,b'}}{\mu_1^{i,b'}+\mu_2^{i,b'}} p_{slash}^{i,b'} \, c_{veg}^{i,b'} \, \delta A_{cover}^{i,b'\to b} + \frac{\mu_2^{i,b}}{\mu_1^{i,b}+\mu_2^{i,b}} p_{slash}^{i,b} \, \delta H_{wood}^{i,b} + \sum_{b'} \frac{\mu_2^{i,b'}}{\mu_1^{i,b'}+\mu_2^{i,b'}} p_{slash}^{i,b'} \, c_{veg}^{i,b'} \, p_{shift}^{i,b'} \, \delta A_{shift}^{i,b'\to b} \tag{30}$$

It should be noted that this initialization step is carbon neutral with respect to the atmosphere:

$$\sum_b \left(\delta C_{bk,veg}^{i,b} + \delta C_{bk,soil1}^{i,b} + \delta C_{bk,soil2}^{i,b} + F_{hwp}^{i,b} + F_{slash1}^{i,b} + F_{slash2}^{i,b}\right) = 0 \tag{31}$$

   Notwithstanding the initialization fluxes, there is a clear similarity between Eq. (20–22) and Eq. (3–5). With the exception of NPP$_{bk}$, all the natural fluxes then follow a similar formulation as Eq. (6–14) for the intensive cycle:

$$NPP_{bk}^{i,b} = 0 \tag{32}$$

$$E_{bk,fire}^{i,b} = \iota^{i,b} \, \mathcal{F}_{fire}^{i,b}(CO_2, T^i, P^i) \, C_{bk,veg}^{i,b} \tag{33}$$

$$E_{bk,harv}^{i,b} = \epsilon_{harv}^{i,b} \, C_{bk,veg}^{i,b} \tag{34}$$

$$E_{bk,graz}^{i,b} = \epsilon_{graz}^{i,b} \, C_{bk,veg}^{i,b} \tag{35}$$

$$F_{bk,mort1}^{i,b} = \mu_1^{i,b} \, C_{bk,veg}^{i,b} \tag{36}$$

$$F_{bk,mort2}^{i,b} = \mu_2^{i,b} \, C_{bk,veg}^{i,b} \tag{37}$$

$$F_{bk,met}^{i,b} = \mu_{met}^{i,b} \, \mathcal{F}_{rh}^{i,b}(T^i, P^i) \, C_{bk,soil1}^{i,b} \tag{38}$$

$$Rh_{bk,1}^{i,b} = \rho_1^{i,b} \, \mathcal{F}_{rh}^{i,b}(T^i, P^i) \, C_{bk,soil1}^{i,b} \tag{39}$$

$$Rh_{bk,2}^{i,b} = \rho_2^{i,b} \, \mathcal{F}_{rh}^{i,b}(T^i, P^i) \, C_{bk,soil2}^{i,b} \tag{40}$$

Recall that the extensive cycle is formulated as a difference to the steady-state that the perturbed ecosystems would reach in an infinite amount of time. Equation (32) therefore means that there is no difference in NPP between undisturbed and disturbed ecosystems, in OSCAR. See Discussion of this model feature. Finally, harvested wood products decay following a 555   product-specific timescale ($\tau_{hwp}$), which leads to carbon emission ($E_{hwp}$):





$$E_{hwp}^{i,b,w} = C_{hwp}^{i,b,w} / \tau_{hwp}^{i,b,w} \tag{41}$$

All the parameters introduced specifically for the extensive cycle follow the definitions and values of earlier versions of OSCAR (Gasser et al., 2017), with two exceptions. First, the above-ground biomass fractions ($\pi_{agb}$) were recalibrated on the DGVMs, along with the intensive cycle parameters. Second, change from v2.3 to v2.4 simplified the treatment of harvested

wood products, but also introduced more uncertainty in their lifetime ($\tau_{hwp}$).

The global land carbon sink ($F_{land}$) is defined as:

$$F_{land} = \sum_{i,b} \left( \left( npp^{i,b} - e_{fire}^{i,b} - e_{harv}^{i,b} - e_{graz}^{i,b} - rh_1^{i,b} - rh_2^{i,b} \right) A_{land}^{i,b} \right) \tag{42}$$

and the emissions caused by land-use and land-cover change ($E_{luc}$) are defined as:

$$E_{luc} = \sum_{i,b} \left( \sum_w E_{hwp}^{i,b,w} + E_{bk,fire}^{i,b} + E_{bk,harv}^{i,b} + E_{bk,graz}^{i,b} + Rh_{bk,1}^{i,b} + Rh_{bk,2}^{i,b} - NPP_{bk}^{i,b} \right) \tag{43}$$

The combination of both equals the net land-to-atmosphere flux, as demonstrated in (Gasser and Ciais, 2013).

### A.8. Detailed calibration protocol

In this section, we make explicit the link between our model's parameters and DGVMs' carbon fluxes and pools, using the standardized CMIP variable names . These variables are: "cLitter" (litter pool), "cRoot" (biomass in root), "cSoil" (soil pool), "cVeg" (vegetation pool), "gpp" (gross primary productivity), "fDOC" (flux of dissolved organic carbon), "fFire"

(wildfire emissions), "fGrazing" (emission from grazing), "fHarvest" (emission from harvested crop products, "fLitterSoil" (flux from litter to soil), "fVegLitter" (flux from vegetation to litter), "fVegSoil" (flux from vegetation to soil), "landCoverFrac" (land-cover fractions), "npp" (net primary productivity), "ra" (autotrophic respiration), and "rh" (heterotrophic respiration).

First, a given DGVM is considered good for emulation if and only if it provides at least the following key variables: npp

(or gpp and ra), rh, cVeg, cSoil and landCoverFrac. Second, the calibration on that model will follow the three-box model iff cLitter, cSoil and fLitterSoil are all provided; it will follow the two-box model otherwise (see Sect. 2.2). Third, an extra variable "grid" is created for each model, corresponding to the area of land in each of the model grid cell, using the land mask provided with the LUH2 land-use and land-cover change data set. Fourth, for each variable and each GCB simulation, the model's spatially explicit data is aggregated into regional and biome-specific time series (i.e. defined on the $i$ and $b$ axes)

using a regional mask ("mask") adapted to the model's resolution, and its own landCoverFrac data aggregated onto OSCAR's biomes. The latter is used to split a variable's value in a given grid cell ($g$) among the model's biomes. For any variable "Var", the resulting time series (along the $t$ axis) follows:

$$\text{Var}(t, i, b) = \frac{\sum_g \text{mask}(g,i) \, \text{Var}(t,g) \, \text{grid}(g) \, \text{landCoverFrac}(t,g,b)^3}{\sum_g \text{landCoverFrac}(t,g,b)^3} \tag{44}$$

The biome area fraction map is taken to the power 3 to give more importance, in a given region, to the grid cells in which

biomes are purer, without taking the risk of excluding any of those grid cells (e.g. by setting a threshold of biome area fraction instead). This processing is also done with Var being an array full of ones, in which case we obtain the "area"





variable corresponding to the area of each biome in each region. Fifth, to correspond to our assumption of a steady-state, the obtained time series are averaged over the whole simulation for S0, and over 1990-2010 for S4.

In the second-to-last step, intermediate variables are defined, with fallback definitions to overcome the unavailability of some DGVMs' outputs:

$$\text{npp}' = \begin{cases} \text{npp}, & \text{if npp exists} \\ \text{gpp} - \text{ra}, & \text{otherwise} \end{cases} \tag{45}$$

$$\text{fMort} = \begin{cases} \text{fVegLitter} + \text{fVegSoil}, & \text{if fVegLitter or fVegSoil exists} \\ \text{npp}' - \text{fFire} - \text{fHarvest} - \text{fGrazing}, & \text{otherwise} \end{cases} \tag{46}$$

$$\text{rh}' = \begin{cases} \text{rh}, & \text{if rh exists} \\ \text{npp}' - \text{fFire} - \text{fHarvest} - \text{fGrazing} - \text{fDOC}, & \text{otherwise} \end{cases} \tag{47}$$

And assuming any other variables' value is zero if it was not reported in the GCB2018 data base, we determine OSCAR's parameters over each pair $(i,b)$ as follows:

$$\eta = \text{npp}'/\text{area} \tag{48}$$

$$\iota = \text{fFire}/\text{cVeg} \tag{49}$$

$$\epsilon_{harv} = \text{fHarvest}/\text{cVeg}, \tag{50}$$

$$\epsilon_{graz} = \text{fGrazing}/\text{cVeg} \tag{51}$$

$$\mu_1 = \begin{cases} \text{fVegLitter}/\text{cVeg}, & \text{if 3 boxes} \\ 0, & \text{otherwise} \end{cases} \tag{52}$$

$$\mu_2 = \begin{cases} \text{fVegSoil}/\text{cVeg}, & \text{if 3 boxes} \\ \text{fMort}/\text{cVeg}, & \text{otherwise} \end{cases} \tag{53}$$

$$\mu_{met} = \begin{cases} \text{fLitterSoil}/\text{cLitter}, & \text{if 3 boxes} \\ 0, & \text{otherwise} \end{cases} \tag{54}$$

$$\rho_1 = \begin{cases} (\text{fVegLitter} - \text{fLitterSoil})/\text{cLitter}, & \text{if 3 boxes} \\ 0, & \text{otherwise} \end{cases} \tag{55}$$

$$\rho_2 = \begin{cases} (\text{fVegSoil} + \text{fLitterSoil})/\text{cSoil}, & \text{if 3 boxes} \\ \text{rh}'/(\text{cLitter} + \text{cSoil}), & \text{otherwise} \end{cases} \tag{56}$$

$$\pi_{agb} = 1 - \text{cRoot}/\text{cVeg} \tag{55}$$

Finally, two ultimate adjustments are made after this whole processing. First, if a DGVM lacks a given biome (such as Cropland, Pasture or Urban), all parameters but $\varepsilon_{harv}$ and $\varepsilon_{graz}$ are assumed the same as those of the Non-Forest biome, with the only exception of Urban $\eta$ that is then zero. Second, $\iota$ is set to zero in the Urban biome, and $\varepsilon_{harv}$ and $\varepsilon_{graz}$ is set to zero in all biomes but Cropland and Pasture, respectively.

### A.9. Processing of LULCC data sets for OSCAR

The "natural land" biome of LUH1 variants had to be split between Forest and Non-Forest. We did so on the data set's own grid, following the potential biomass map provided with the LUH1 data set, and a threshold value given by (Hurtt et al., 2006) of 2 kgC m$^{-2}$ above which a grid cell's natural land was considered 100% Forest, and below which it was split between





Forest and Non-Forest. This split was made such that the proportion of Forest equals the potential biomass divided by the
threshold value. In addition, shifting cultivation transitions were calculated by isolating all reciprocal transitions between
natural land and anthropogenic biomes within the shifting cultivation mask provided with the LUH1 data set. Note that the
LUH1-GCB2015 data set does not include urban land.

The natural biomes of the LUH2 variants match those of OSCAR. The anthropogenic biomes, however, are more finely
defined than in our model (i.e. more cropland and pasture types). We therefore aggregated all the cropland types into one
unique Cropland biome, and similarly with pasture types. Worthy of note is the fact that we assume rangeland to be pastures,
which may explain some of the differences shown in Figure A2. Additionally, following information provided by the LUH
team, shifting cultivation transitions were calculated by isolating reciprocal transitions between any of the two natural
biomes and cropland, and only between the 33°N and 33°S latitudes.

The FRA2015 data as used by the H&N model of (Houghton and Nassikas, 2017) demanded little processing to be
compatible with OSCAR. As stated in Discussion, forest plantations were assimilated to natural forests. The two types of
harvested wood (fuel and industrial) were summed together, which by construction leads to a split between HWP pools that
is different from that of H&N. We created our shifting cultivation driver by doing the cumulative sum of the yearly
transitions towards what they identified as being newly established shifting cultivation areas. This was taken in tropical
countries only, and divided by 15 years to assume a 15-year rotation time following (Hurtt et al., 2011). No urban biome is
included in this data set.



**Acknowledgements**

We thank all the organizers and contributors to the TRENDY modeling exercise, as their work was key to recalibrating OSCAR. TG and YQ acknowledge support from the European Research Council Synergy project "Imbalance-P" (grant ERC-2013-SyG-610028).

**Author contribution**

TG designed the study. TG developed all successive versions of OSCAR and its bookkeeping module. RAH provided FRA2015 input data. YQ and TG formatted all input data for OSCAR. LC processed the TRENDYv7 data and recalibrated OSCAR, under supervision of YQ and TG. Simulations were setup by LC and TG. TG drafted the manuscript. All authors contributed to the final analysis and to the manuscript.

**Competing interests**

The authors declare that they have no conflict of interest.

**Code and data availability**

Our estimates of LULCC emissions, the land sink, the LASC, and their respective breakdown will be made openly available online upon acceptance of the paper. The source code of OSCAR is available at https://github.com/tgasser/OSCAR. Additional scripts are available upon request to the corresponding author.





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





**Table 1. Availability of LULCC emissions estimates in the GCB2019 and this study.** This follows our three main axes of analysis: definition (axis *i*), driving data sets (*ii*) and biogeochemical parameterization (*iii*).

| | GCB2019 (Friedlingstein et al., 2019) | | | | OSCAR (this study) | | | |
|---|---|---|---|---|---|---|---|---|
| *(i) Definition (→)* | excl. LASC | | incl. LASC | | excl. LASC | | incl. LASC | |
| *(ii) LULCC data set (→)* | LUH | FRA | LUH | FRA | LUH | FRA | LUH | FRA |
| *(iii) Biogeochemical parameters[†] (↓)* | | | | | | | | |
| BLUE | X | | | | | | | |
| H&N | | X | | | | | | |
| CABLE-POP | | | X | | X | X | X | X |
| CLASS-CTEM | | | X | | X | X | X | X |
| CLM5.0[‡] | | | X | | | | | |
| DLEM | | | X | | X | X | X | X |
| ISAM | | | X | | X | X | X | X |
| JSBACH | | | X | | X | X | X | X |
| JULES | | | X | | X | X | X | X |
| LPJ[‡] | | | X | | | | | |
| LPJ-GUESS | | | X | | X | X | X | X |
| LPX-Bern[‡] | | | X | | | | | |
| OCN | | | X | | X | X | X | X |
| ORCHIDEE | | | X | | X | X | X | X |
| ORCHIDEE-CNP | | | X | | X | X | X | X |
| SDGVM[‡] | | | X | | | | | |
| SURFEX[‡] | | | X | | | | | |
| VISIT | | | X | | X | X | X | X |

[†] BLUE (Hansis et al., 2015) and H&N (Houghton and Nassikas, 2017) are bookkeeping models; others are DGVMs.

[‡] OSCAR could not be calibrated on these DGVMs because of insufficient data.





**Table 2. Estimates of the global net land-to-atmosphere flux, LULCC emissions, land sink and LASC.** Estimates following our
785  default and alternative constraints are provided. The land sink includes the LASC, and therefore the net land-to-atmosphere flux is strictly
equal to LULCC emissions minus the land sink.

|  | annual flux (PgC yr[-1]) | | cumulative flux (PgC) | |
| --- | --- | --- | --- | --- |
|  | 2018 | 2009–2018 | 1850–2018 | 1750–2018 |
| **Default constraint (net land flux as residual from fossil emissions, atmospheric growth and ocean sink; best guess)** | | | | |
| Net land-to-atmosphere flux[†] | −1.85±0.75 | −1.62±0.79 | −27±26 | −22±29 |
| Bookkeeping LULCC emissions | 1.39±0.43 | 1.36±0.42 | 178±50 | 206±57 |
| Land carbon sink | 3.24±1.02 | 2.98±1.02 | 205±53 | 228±59 |
| Loss of additional sink capacity | 0.78±0.62 | 0.68±0.57 | 31±22 | 32±23 |
| **Alternative constraint (land sink without LULCC perturbation as estimated by the DGVMs)** | | | | |
| Net land-to-atmosphere flux[†] | −1.51±0.66 | −1.33±0.71 | −16±47 | −9±54 |
| Bookkeeping LULCC emissions | 1.27±0.36 | 1.26±0.36 | 166±44 | 192±51 |
| Land carbon sink | 2.78±0.68 | 2.58±0.73 | 181±36 | 201±40 |
| Loss of additional sink capacity | 0.51±0.33 | 0.44±0.32 | 21±11 | 22±11 |

[†] Counted algebraically: negative values denote carbon removal from the atmosphere.





790 **Table 3. Uncertainty sources in our estimates of bookkeeping LULCC emissions and LASC.** They are expressed as debiased standard deviation (1-σ) and coefficients of variation (CV; in parenthesis).

| | annual flux (PgC yr$^{-1}$) | | cumulative flux (PgC) | |
|---|---|---|---|---|
| | 1995–2004 | 2009–2018 | 1850–2004 | 1750–2018 |
| **Uncertainty breakdown of bookkeeping LULCC emissions** | | | | |
| (i) Definition | ±0.31 (21%) | ±0.43 (25%) | ±14 (8%) | ±20 (9%) |
| (ii) LULCC data set | ±0.30 (24%) | ±0.03 (2%) | ±8 (5%) | ±8 (4%) |
| *-- LUH data set version* | *±0.14 (14%)* | *--* | *±14 (8%)* | *--* |
| *-- FRA data set version*[†] | *±0.25 (21%)* | *--* | *±15 (10%)* | *--* |
| (iii) Biogeochemical parameters | ±0.40 (32%) | ±0.40 (29%) | ±43 (27%) | ±55 (27%) |
| *-- only HWP-related parameters* | *±0.03 (2%)* | *±0.02 (2%)* | *±4 (2%)* | *±5 (3%)* |
| **Uncertainty breakdown of the loss of additional sink capacity** | | | | |
| (ii) LULCC data set | ±0.14 (28%) | ±0.21 (31%) | ±7 (31%) | ±10 (31%) |
| (iii) Biogeochemical parameters | ±0.35 (75%) | ±0.50 (77%) | ±13 (62%) | ±19 (63%) |

[†] These values are taken directly from (Houghton and Nassikas, 2017) and were therefore not computed with OSCAR. They include some biogeochemical uncertainty to a lesser but unknown degree.





**Table 4. Regional breakdown of bookkeeping LULCC emissions and LASC.** This is provided for our best-guess estimates and the two main LULCC data sets (LUH2-GCB2019 and FRA2015) separately. Regions are defined in (Houghton and Nassikas, 2017).

| | annual flux 2009–2018 (PgC yr$^{-1}$) | | | cumulative flux 1750–2018 (PgC) | | |
|---|---|---|---|---|---|---|
| | best guess | LUH | FRA | best guess | LUH | FRA |
| **Bookkeeping LULCC emissions (ELUC)** | | | | | | |
| Sub–Saharan Africa | 0.46±0.24 | 0.29±0.11 | 0.66±0.20 | 28±13 | 25±14 | 31±9 |
| Latin America | 0.63±0.23 | 0.52±0.14 | 0.76±0.24 | 63±18 | 67±18 | 59±17 |
| South and Southeast Asia | 0.32±0.11 | 0.35±0.09 | 0.29±0.12 | 39±12 | 36±8 | 42±14 |
| North America | 0.00±0.04 | 0.02±0.03 | −0.02±0.03 | 34±13 | 34±14 | 35±13 |
| Europe | −0.03±0.03 | −0.02±0.02 | −0.05±0.03 | 2±3 | 4±2 | −1±2 |
| Former Soviet Union | 0.01±0.05 | 0.03±0.03 | −0.02±0.05 | 20±12 | 20±12 | 20±12 |
| China | −0.05±0.21 | 0.14±0.07 | −0.27±0.05 | 13±13 | 23±8 | 1±5 |
| North Africa and the Middle East | −0.01±0.01 | 0.00±0.01 | −0.01±0.01 | −1±2 | −1±2 | 0±2 |
| East Asia | 0.00±0.05 | 0.01±0.04 | −0.01±0.01 | 3±2 | 4±2 | ~1 |
| Oceania | 0.03±0.06 | 0.00±0.04 | 0.07±0.07 | 6±10 | 1±8 | 11±10 |
| **Loss of additional sink capacity (LASC)** | | | | | | |
| Sub–Saharan Africa | 0.12±0.13 | 0.16±0.17 | 0.08±0.04 | 5±6 | 7±7 | 2±1 |
| Latin America | 0.18±0.18 | 0.21±0.21 | 0.15±0.14 | 9±5 | 10±6 | 7±4 |
| South and Southeast Asia | 0.11±0.08 | 0.11±0.08 | 0.11±0.07 | 4±2 | 4±3 | 4±2 |
| North America | 0.10±0.08 | 0.11±0.09 | 0.08±0.07 | 5±4 | 6±5 | 5±4 |
| Europe | 0.01±0.02 | 0.03±0.02 | 0.00±0.01 | 1±1 | 2±1 | ~0 |
| Former Soviet Union | 0.06±0.06 | 0.06±0.06 | 0.05±0.05 | 2±3 | 3±3 | 2±2 |
| China | 0.07±0.10 | 0.12±0.10 | 0.02±0.07 | 4±4 | 5±4 | 2±3 |
| North Africa and the Middle East | 0.00±0.01 | 0.00±0.01 | 0.00±0.01 | ~0 | ~0 | ~0 |
| East Asia | 0.01±0.01 | 0.02±0.01 | ~0.00 | 1±1 | 1±1 | ~0 |
| Oceania | 0.02±0.11 | 0.02±0.14 | 0.03±0.05 | 1±2 | 1±3 | 1±2 |





**Table 5. Breakdown of bookkeeping LULCC emissions and LASC by LULCC activity.** This provided for our best-guess estimates and the two main LULCC data sets (LUH2-GCB2019 and FRA2015) separately.

| | annual flux 2009–2018 (PgC yr$^{-1}$) | | | cumulative flux 1750–2018 (PgC) | | |
|---|---|---|---|---|---|---|
| | best guess | LUH | FRA | best guess | LUH | FRA |
| **Bookkeeping LULCC emissions (ELUC)** | | | | | | |
| Deforestation for cropland | 1.86±0.57 | 2.20±0.48 | 1.47±0.37 | 213±93 | 285±62 | 131±38 |
| Other deforestation | 0.55±0.26 | 0.41±0.14 | 0.70±0.27 | 77±27 | 88±26 | 64±24 |
| Reforestation and afforestation | −1.36±0.49 | −1.70±0.38 | −0.96±0.22 | −144±99 | −230±48 | −45±11 |
| Other natural land appropriation | 0.42±0.31 | 0.63±0.26 | 0.17±0.08 | 60±33 | 82±30 | 35±12 |
| Other natural land (re)establishment | −0.21±0.18 | −0.33±0.15 | −0.07±0.08 | −20±18 | −34±11 | −3±2 |
| Conversions among anthrop. biomes | 0.02±0.02 | 0.03±0.02 | 0.01±0.01 | 1±1 | 1±1 | ~0 |
| Wood harvest | 0.09±0.04 | 0.10±0.04 | 0.07±0.03 | 19±6 | 21±5 | 16±6 |
| **Loss of additional sink capacity (LASC)** | | | | | | |
| Deforestation for cropland | 0.29±0.16 | 0.31±0.17 | 0.27±0.15 | 14±6 | 16±7 | 12±5 |
| Other deforestation | 0.24±0.15 | 0.30±0.15 | 0.17±0.10 | 12±6 | 15±6 | 7±3 |
| Reforestation and afforestation | −0.15±0.10 | −0.21±0.10 | −0.09±0.04 | −7±5 | −10±4 | −3±1 |
| Other natural land appropriation | 0.39±0.53 | 0.57±0.63 | 0.19±0.24 | 16±21 | 24±25 | 8±9 |
| Other natural land (re)establishment | −0.09±0.12 | −0.14±0.14 | −0.03±0.06 | −3±4 | −5±5 | −1±1 |
| Conversions among anthrop. biomes | 0.00±0.01 | 0.00±0.01 | ~0.00 | ~0 | ~0 | ~0 |
| Wood harvest | 0.00 | 0.00 | 0.00 | 0 | 0 | 0 |





Figure 1. **Our best-guess estimates of bookkeeping LULCC emissions and LASC.** Panel (a) shows annual fluxes, and panel (b) shows cumulative ones. The net land-to-atmosphere flux is also shown in panel (b) and compared to the constraint (red). Shaded areas show the 1-σ uncertainty range. Panel (c) shows the detailed probability distributions of the cumulative net land flux, land sink, and LULCC emissions, in the unconstrained (dotted histograms) and constrained (plain ones) output ensemble (20,000 Monte Carlo elements), compared to the constraint (red) and the GCB estimates (dashed black).





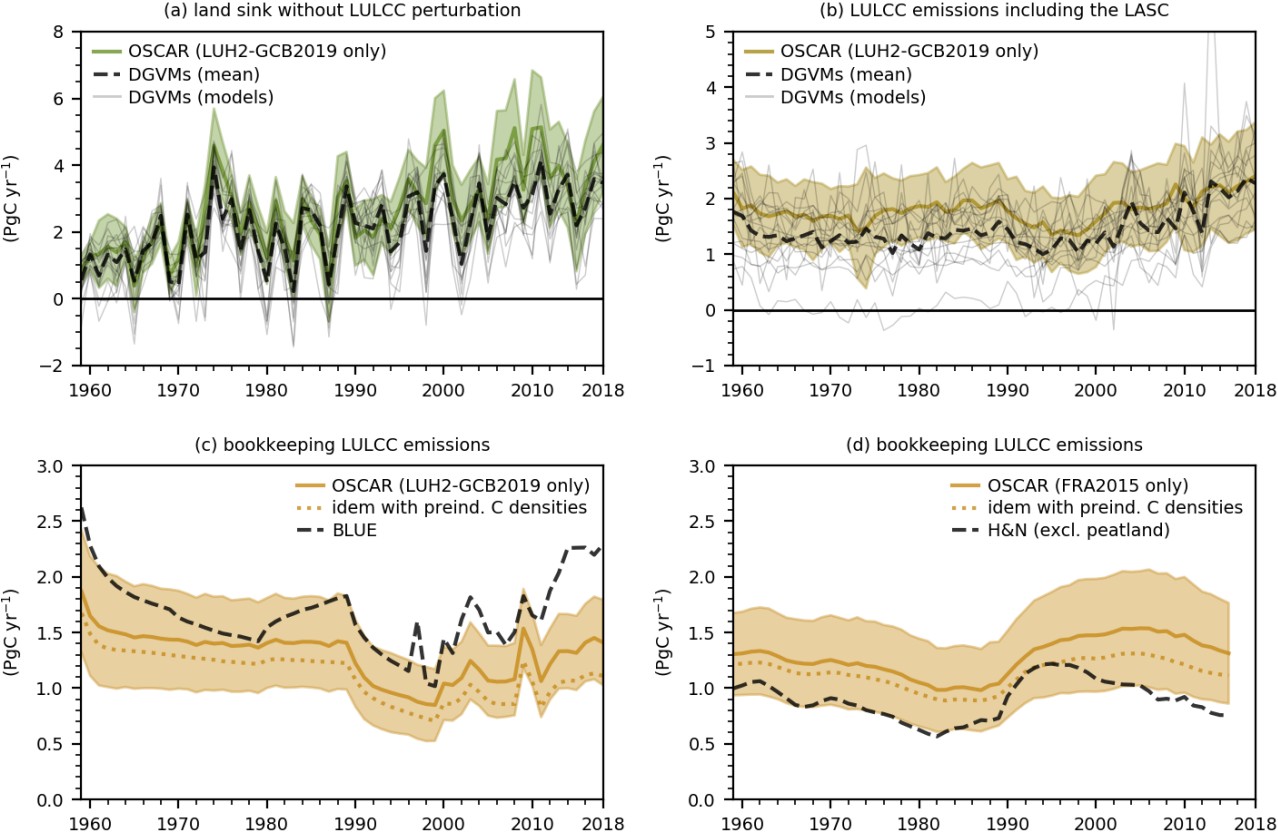

**Figure 2. Comparison of our results to the GCB2019.** (a) Annual terrestrial carbon sink in the absence of LULCC perturbation simulated by OSCAR (color), the individual GCB DGVMs (light grey) and their multi-model mean (dashed black) (b) Annual LULCC emissions deduced from the GCB DGVMs (i.e. including the LASC). (c) Bookkeeping LULCC emissions when the model is driven by the LUH2-GCB2019 data set, compared to BLUE estimates reported by the GCB2019. Dotted line shows the same emissions but when carbon densities are kept at their preindustrial throughout the simulation, and without uncertainty for legibility. (d) Bookkeeping LULCC emissions when the model is driven by the FRA2015 data set, compared to H&N estimates from which emissions from peatlands were subtracted. All shaded areas show the 1-σ uncertainty range.





**Figure 3. Variations and uncertainties in time-series of global annual LULCC emissions.** (a) Effect of adding the LASC to LULCC emissions. (b) Effect of the LULCC driving data sets (only the two data sets used to estimate our best guess). (c) Effect of the data set version among LUH variants (not used for our best guess). (d) Effect of the data set version among FRA variants. These emissions were not simulated by OSCAR; they were reported by (Houghton and Nassikas, 2017) (their Figure 8). (e) Effect of all the parameters of OSCAR (using the weighted Monte Carlo ensemble). (f) Effect of the subset of parameters of OSCAR that are related to harvested wood products (i.e. the parameters that are not derived from DGVMs). All panels show bookkeeping LULCC emissions, with the obvious exception of panel (a). Thick colored lines show the values obtained by averaging over all axes of analysis other than the one investigated in the panel. Dashed grey lines with markers show the debiased coefficients of variation (CV), that is the ratio of the debiased standard deviation over the average, and refer to the y-axis on the right-hand side of each panel.



**Figure 4. Regional breakdown of our best-guess estimates.** Annual bookkeeping LULCC emissions (in brown) and LASC (in green) are shown, except in the last two panels where the regional cumulative bookkeeping LULCC emissions (ELUC) and LASC over 1750–2018 are shown. Shaded areas and uncertainty bars represent the 1-σ uncertainty range. To help identify regional discrepancies between LULCC driving data sets, we also separate the average estimates for the LUH2-GCB2019 (dashed black line) and FRA2015 (dotted black line) data sets, without their own uncertainty for legibility.





**Figure 5. Breakdown of our best-guess estimates by LULCC activities.** Annual bookkeeping LULCC emissions (in brown) and LASC (in green) are shown, except in the last two panels that show the regional cumulative bookkeeping LULCC emissions (ELUC) and LASC over 1750–2018. Shaded areas and uncertainty bars represent the 1-σ uncertainty range. Similarly to Figure 3, we also separate the average estimates for the LUH2-GCB2019 (dashed black line) and FRA2015 (dotted black line) data sets, without their own uncertainty for legibility.





840

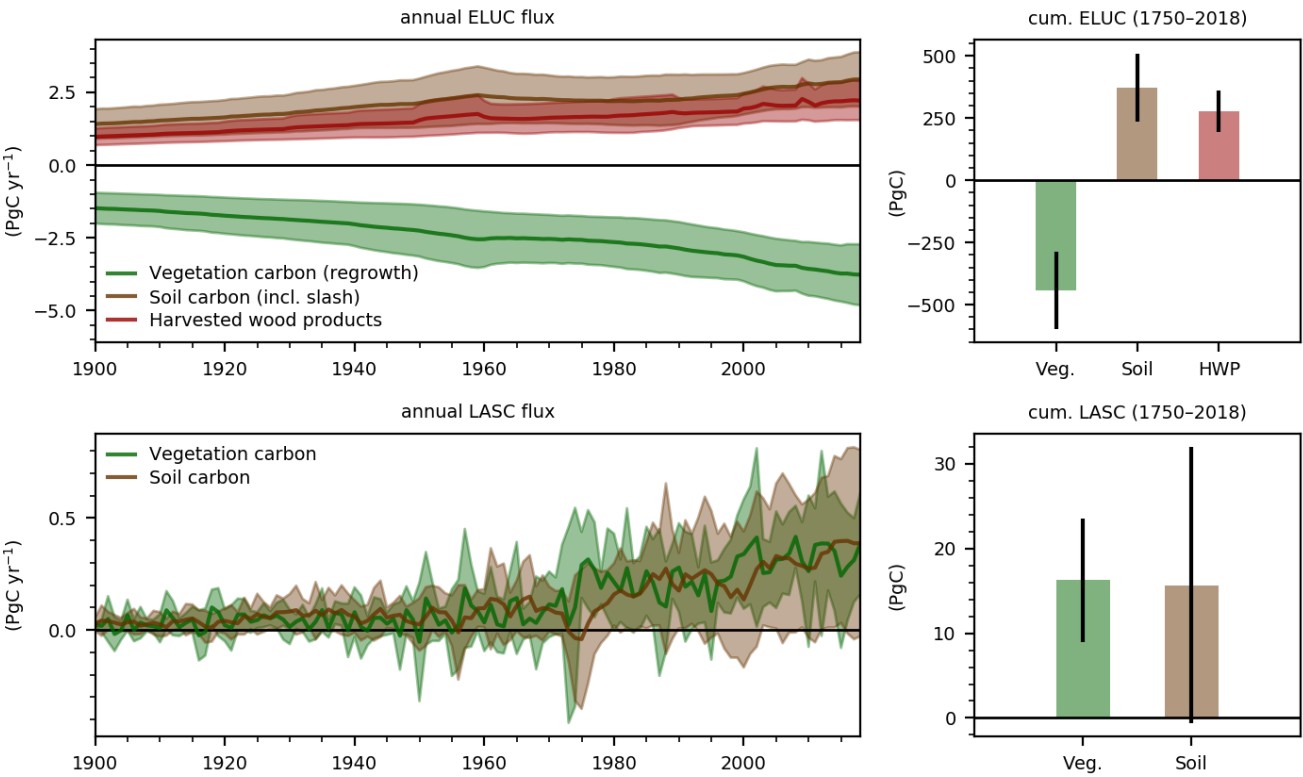

**Figure 6. Breakdown of our best-guess estimates by carbon pools:** vegetation (green), soils (brown), and HWPs (red). Shaded areas and uncertainty bars represent the 1-σ uncertainty range.





**Table A1. Preindustrial NPP and carbon stocks in GCB2018 DGVMs and in their emulation by OSCAR.** The "DGVM" columns
845 show the carbon fluxes or stocks we extracted for the calibration of OSCAR. Because we used our own land area map and regional mask
to process the original DGVM outputs, and then aggregated regional values into global values, these values may slightly differ from a
direct extraction of the DGVM outputs. The GCB2018 protocol did not require the DGVM teams to provide their own land area map.

| | NPP (PgC yr$^{-1}$) | | | $C_{veg}$ (PgC) | | | $C_{soil}$ (PgC) | | |
|---|---|---|---|---|---|---|---|---|---|
| *Carbon densities:* | OSCAR | | DGVM | OSCAR | | DGVM | OSCAR | | DGVM |
| *Land-cover (1700):* | LUH2 | DGVM | DGVM | LUH2 | DGVM | DGVM | LUH2 | DGVM | DGVM |
| CABLE-POP | 43.0 | 43.0 | 43.0 | 456 | 474 | 506 | 1441 | 1456 | 1541 |
| CLASS-CTEM | 45.4 | 45.5 | 45.9 | 384 | 402 | 410 | 1050 | 1049 | 1068 |
| DLEM | 51.4 | 54.4 | 54.4 | 412 | 457 | 486 | 1047 | 1105 | 1151 |
| ISAM | 50.4 | 46.4 | 46.9 | 598 | 505 | 558 | 958 | 954 | 1042 |
| JSBACH | 50.4 | 45.9 | 46.6 | 416 | 380 | 365 | 714 | 662 | 670 |
| JULES | 56.2 | 53.1 | 53.2 | 580 | 529 | 561 | 1341 | 1269 | 1350 |
| LPJ-GUESS | 50.9 | 49.1 | 49.3 | 422 | 396 | 404 | 1385 | 1337 | 1332 |
| OCN | 55.5 | 57.3 | 57.6 | 471 | 531 | 566 | 1619 | 1665 | 1754 |
| ORCHIDEE | 41.5 | 43.3 | 43.6 | 316 | 348 | 360 | 627 | 647 | 671 |
| ORCHIDEE-CNP | 42.2 | 43.4 | 43.8 | 329 | 356 | 380 | 690 | 699 | 740 |
| VISIT | 45.4 | 46.7 | 46.4 | 415 | 434 | 435 | 1208 | 1244 | 1218 |
| mean | 48.4 | 48.0 | 48.2 | 436 | 437 | 457 | 1098 | 1099 | 1140 |
| std | 4.9 | 4.7 | 4.6 | 85 | 63 | 77 | 317 | 321 | 336 |





**Table A2. Global preindustrial (1700) land-cover in the GCB2018 DGVMs, compared to that in our two main LULCC data sets.**

|  | Forest | Non-Forest | Cropland | Pasture | Urban |
|---|---|---|---|---|---|
| CABLE-POP | 4876 | 7619 | -- | -- | -- |
| CLASS-CTEM | 5465 | 6579 | 276 | -- | -- |
| DLEM | 5129 | 7435 | 278 | 141 | 1 |
| ISAM | 3205 | 8824 | 257 | 455 | -- |
| JSBACH | 4144 | 7555 | 267 | 236 | -- |
| JULES | 3809 | 9026 | -- | -- | 18 |
| LPJ-GUESS | 3769 | 8511 | 279 | 432 | -- |
| OCN | 5982 | 6488 | 296 | -- | -- |
| ORCHIDEE | 5224 | 7374 | 387 | -- | -- |
| ORCHIDEE-CNP | 5224 | 7020 | 387 | 361 | -- |
| VISIT | 5278 | 7176 | 275 | -- | -- |
| mean | 4737 | 7601 | 300 | 325 | 9 |
| std | 826 | 811 | 47 | 119 | 8 |
| LUH2-GCB2019 | 4620 | 7256 | 357 | 756 | 1 |
| FRA2015 | 5013 | 6074 | 574 | 1756 | -- |

850



**Table A3. Comparison of regional bookkeeping LULCC emissions to H&N estimates.** H&N values are taken directly from (Houghton and Nassikas, 2017): their annual flux is over 2006–2015, and their cumulative flux over 1850–2015.

| | best guess | LUH2-GCB2019 | FRA2015 | H&N |
|---|---|---|---|---|
| | *annual flux 2009–2018 (TgC yr$^{-1}$)* | | | |
| Sub−Saharan Africa | 460±243 | 290±110 | 656±201 | *437±55* |
| Latin America | 632±227 | 518±136 | 762±238 | *527±114* |
| South and Southeast Asia | 321±111 | 349±89 | 290±120 | *441±141* |
| North America | 1±37 | 20±26 | −22±34 | *−73±79* |
| Europe | −34±30 | −17±21 | −54±25 | *−102±46* |
| Former Soviet Union | 8±48 | 33±34 | −20±46 | *−60±55* |
| China | −53±213 | 138±71 | −271±53 | *−58±112* |
| North Africa and the Middle East | −8±9 | −3±6 | −13±8 | *−3±55* |
| East Asia | 0±48 | 11±41 | −11±8 | *−3* |
| Oceania | 34±64 | 4±36 | 69±71 | *8* |
| | *cumulative flux 1850–2018 (PgC)* | | | |
| Sub−Saharan Africa | 26.8±11.8 | 23.1±12.6 | 31.0±9.3 | *24.1±3.0* |
| Latin America | 60.6±17.3 | 64.0±17.4 | 56.6±16.3 | *37.5±3.4* |
| South and Southeast Asia | 34.3±11.3 | 30.1±7.2 | 39.3±13.1 | *40.5±7.8* |
| North America | 29.0±11.4 | 30.3±12.1 | 27.5±10.3 | *22.7±6.3* |
| Europe | −1.0±3.7 | 2.0±1.5 | −4.4±2.2 | *−5.2±3.7* |
| Former Soviet Union | 15.2±10.1 | 15.5±10.1 | 15.0±10.0 | *10.7±4.3* |
| China | 7.4±9.9 | 15.5±5.5 | −1.8±4.1 | *7.3±7.0* |
| North Africa and the Middle East | −0.9±1.4 | −1.2±1.6 | −0.6±1.2 | *2.7±6.6* |
| East Asia | 1.9±1.5 | 3.0±1.2 | 0.6±0.4 | *1.4* |
| Oceania | 5.1±9.2 | 1.0±7.4 | 9.7±8.9 | *3.9* |





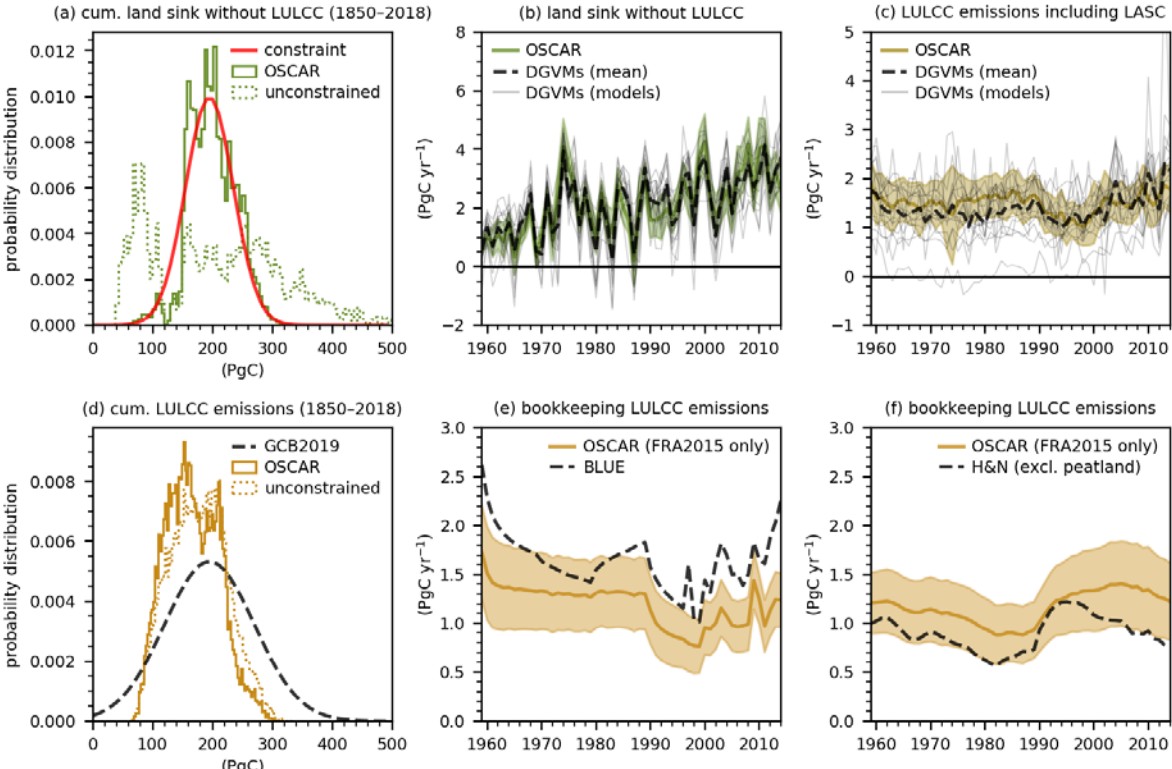

**Figure A1. Effect of the alternative constraint and comparison to the GCB2019.** (a) Probability distributions in the unconstrained (dotted histograms) and constrained (plain ones) OSCAR ensembles of the cumulative terrestrial carbon sink over 1850–2018 in the absence of LULCC perturbation, compared to the constraint (red). (b) Annual terrestrial carbon sink in the absence of LULCC perturbation simulated by OSCAR (colour), the individual GCB DGVMs (light grey) and their multi-model mean (dashed black). (c) Annual LULCC emissions deduced from the GCB DGVMs (i.e. including the LASC). (d) Probability distributions of our best-guess estimate of the cumulative LULCC emissions over 1850–2018, compared to the GCB estimate (dashed black). (e) Bookkeeping LULCC emissions when the model is driven by the LUH2-GCB2019 data set, compared to BLUE estimates reported by the GCB2019. (f) Bookkeeping LULCC emissions when the model is driven by the FRA2015 data set, compared to H&N estimates from which emissions from peatlands were subtracted. All shaded areas show the 1-σ uncertainty range.





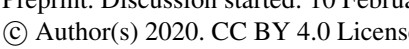

**Figure A2. Summary of LULCC drivers in LUH2-GCB2019 (dashed black lines) and FRA2015 (dotted grey lines).** For each region (along the rows), the first five columns show net are changes in our five biomes (in order: Forest, Non-Forest, Cropland, Pasture, Urban), the second-to-last column shows wood harvest, and the final column shows total area under shifting cultivation.



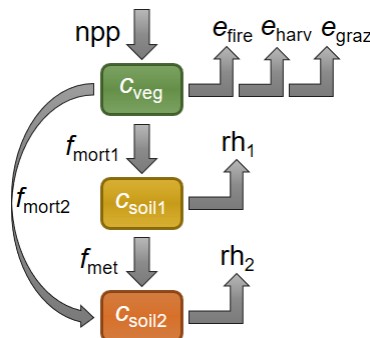

870    **Figure A3. Diagram of the three-box model describing the intensive land carbon cycle in each (region, biome) set.** The three boxes correspond to carbon pools in the vegetation ($c_{veg}$), the litter ($c_{soil1}$) and the soil ($c_{soil2}$). Detailed definitions and formulations of the fluxes (grey arrows) are provided in Appendix A7.