# Peer review of "Historical CO2 emissions from land-use and land-cover change and their uncertainty"

_Biogeosciences, 2020_

## Referee Comment (RC1) · Anonymous Referee #1 · 21 Mar 2020

General points

In this study, the authors investigated the historical carbon emissions caused by land-use change using a reduced-form Earth system model, OSCAR. They conducted a series of ensemble simulations to obtain the best guess and its associated uncertainty of the land-use-induced emission. The estimated historical cumulative emission, 206 $\pm$ 57 Pg C, is substantial and looks consistent with those obtained by previous global carbon budget studies. Land-use change is an important anthropogenic CO2 source and related to various human activities such as agriculture and urbanization. Therefore, clearly, this study falls within the journal scope and will carry implications on the global carbon budget.

On the other hand, the methodology they adopted is slightly complicated. They proposed a unifying approach for the bookkeeping model and dynamic global vegetation models, but I could not understand how these approaches were integrated into the OS-CAR model. It was impressive for me that the model allowed a large number (10,000) of ensemble simulations, but how biogeochemical parameterizations were perturbed was not adequately described. Although the authors provided long appendix, the methodology should be clarified in the main text (section 2.).

The evaluation of the loss of additional sink capacity (LASC) is the remarkable feature of this study, although it looks to depend heavily on previous studies such as GCP2019 and FRA2015. Overall, the manuscript is well-written and I recommend a few amendments as seen below.

Specific points

Line 79: Please provide more specifications of the OSCAR model, such as spatial resolution, spin-up method, time step, etc.

Line 83: Can you explain more about the "10,000 different biogeochemical parameterizations"?

Line 92: In general, Results section should present exclusively the outcomes obtained in the present study and so should not include citations to other studies. The present "3. Results" section looks more like a "Results and Discussion" section. Please consider restructuring of the manuscript.

Line 107: Can you specify what is "the change in empirical constraint" responsible for the larger LASC?

Line 223: Please explain what are the "seven categories of LUCCC activities", in a consistent manner with those in the 2. Overview of the methodology section (only three activities in Line 69–70).

---

## Referee Comment (RC2) · Anna Harper (Referee) · 21 Apr 2020

CO2 emissions from land use and land cover change (LULCC) are uncertain as they are not directly observable. In the annually updated Global Carbon Budget, two methods are used to estimate the emissions: a book-keeping approach based on LULCC data and empirical response functions, and results from process-based dynamic global vegetation models. T. Gasser and co-authors present results from a model (OSCAR) which combines book-keeping approaches with process-based modelling. There are several benefits to this approach, including an additional, constrained estimate of annual CO2 emissions from the land and a method for evaluating sources of uncertainty. This manuscript describes some of these results. Overall I think this is a very strong manuscript, it is well written and forms an important contribution to the Global Carbon

Project. I appreciated having an explanation of the model in the Appendix (although I have noted a few questions about this below). My primary concern is about the choice of constraint and sensitivity of model results to the choice, which I explain below. There a couple of other 'substantial' comments, and a few minor comments.

The constraint:

First, what is the constraint used in reducing the ensemble of simulations (as in, what is the value)? I see it is given at Line 112, but I think this is an important piece of information to include in the Methods and in Section A.5.

The choice of constraint seems very important. It's not clear how robust the results are to the choice of constraint (which is important given how uncertain the constraint is). Although this is not discussed in the manuscript, some information about sensitivity of results to the constraint is provided within the figures and tables. From examining Table 2 – in general there is overlap between the annual and cumulative emissions when comparing the two methods of constraining OSCAR. The biggest differences are for cumulative land carbon sink (which I think makes sense given that the original constraint mostly impacted the land carbon sink in Figure 1c, indicating this is a large source of potential uncertainty), and for LASC (which has a strong dependence on the land carbon sink, so that makes sense as well). I think the manuscript should include discussion of this sensitivity; and justification for choosing the constraint in this study.

A related question I have (although maybe not for this paper): The uncertainty analysis revealed that biogeochemical parameters contribute to large proportion of the model uncertainty. This is attributed to the carbon densities (Lines 175-176), so it makes me think an additional constraint could be carbon stored in soils and vegetation. Have the authors considered using present-day carbon stocks as an additional constraint?

Other comments:

Lines 435-438: The pre-industrial steady state carbon pools are determined using

average climate variables from 1901-1930. I'm not sure how much IAV occurred during this time period, but using average climate, rather than looping through years (as in GCB), will neglect the impact of IAV on the pools. Is it possible this has an impact on the steady state pool values, and would cause spurious transient responses when switching to the historical transient simulations?

Lines 284-286: Annual emissions would be less without changes in environmental conditions (according to the model): in terms of what the model is simulating, why is this? (e.g. is it increasing fire emissions? More land carbon without the changes resulting in higher emissions? Etc)

Minor comments

OSCAR calculates carbon stocks and fluxes for average biomes within 5 regions – what are the biomes? I think these should be stated up front in Section 2 and in A.1. Also state number of regions.

Line 181: Typo; I think you mean Table 3.

Lines 358-359: Could you clarify what "definition 3" and "definition B" are?

Line 383: Recommend replacing 'a fortiori', I had to look it up

Line 439: The word "past" here is a little vague; in my mind it can mean either 'beyond 1950' or 'further in the past than 1950'; perhaps just change to 'after' (I think that's what is meant here).

Lines 435-455: Perhaps reference Figure 1 and related discussion here to note that there is further discussion of the effect of the constraint in the main text.

Lines 451-453: It would be helpful if you state here the number of model parameters, and refer to section A.7 where they are introduced.

Section A.7: Which CMIP5 experiments were used for calibrating the sensitivity functions?

Eq 19: How is tau_shift determined? Also it's not clear how Eq. 13 is used to derive Eq. 19. Is this a typo?

I can't see where Fslash comes into the overall equations, other than the initialization step in Eq. 31.

Sorry if I have missed it, but I also don't see an explanation of how the LASC is calculated. I assume it's related to Eq. 42 but without transient LULCC?

Table A2: What are the units (Mha?)?

Figure A1: Is the brown line in panel (e) mislabelled?

---

## Author Comment (AC1) · 26 May 2020

Comment 1.1

General points

In this study, the authors investigated the historical carbon emissions caused by land-use change using a reduced-form Earth system model, OSCAR. They conducted a series of ensemble simulations to obtain the best guess and its associated uncertainty of the land-use-induced emission. The estimated historical cumulative emission, 206±57 Pg C, is substantial and looks consistent with those obtained by previous global carbon budget studies. Land-use change is an important anthropogenic CO2 source and related to various human activities such as agriculture and urbanization. Therefore,

clearly, this study falls within the journal scope and will carry implications on the global carbon budget.

Response 1.1

We thank the referee their comments and for recognizing the potential impact of our work.

Comment 1.2

On the other hand, the methodology they adopted is slightly complicated. They proposed a unifying approach for the bookkeeping model and dynamic global vegetation models, but I could not understand how these approaches were integrated into the OSCAR model. It was impressive for me that the model allowed a large number (10,000) of ensemble simulations, but how biogeochemical parameterizations were perturbed was not adequately described. Although the authors provided long appendix, the methodology should be clarified in the main text (section 2.).

Response 1.2

Done. The following text was added to section 2.

*These parameterizations are drawn randomly and with equiprobability from a pool of potential sets of parameters. This main pool is obtained by combining smaller pools of available parameterizations for separate processes (or group of processes), as described by (Gasser et al., 2017). For instance, recalibration of the preindustrial steady-state led to 11 possible parameterizations for preindustrial net primary productivity and turnover times, 4 for preindustrial wildfire rates, 5 for preindustrial export fractions from crop harvesting, and 2 for those from animal grazing. This is already a total of $11 \times 4 \times 5 \times 2 = 440$ parameterizations. These are further combined with available parameterizations for other elements such as the transient response of the land carbon cycle to atmospheric CO2 and climate change, or the handling of harvested wood products, which leads to a main pool of $\sim 1.5$ million sets of parameters.*

Comment 1.3

The evaluation of the loss of additional sink capacity (LASC) is the remarkable feature of this study, although it looks to depend heavily on previous studies such as GCP2019 and FRA2015. Overall, the manuscript is well written and I recommend a few amendments as seen below.

Response 1.3

Thank you for the support. We note that our study uses GCB2019 and FRA2015 data as input, and so it does depend on that data as much as any other modelling study depends on their own input data. We do not see that as a weakness.

Comment 1.4

Specific points

Line 79: Please provide more specifications of the OSCAR model, such as spatial resolution, spin-up method, time step, etc.

Response 1.4

The first paragraph of section 2 has been extended with this information: OSCAR is not spatially explicit, but the global land C cycle is divided into 10 broad world regions; it does not require spin-up because the preindustrial steady-state is directly calibrated on TRENDY models; it works with a yearly time-step.

Further information is available in Appendix, and ultimately in the description paper of the model (Gasser et al., 2017).

Comment 1.5

Line 83: Can you explain more about the "10,000 different biogeochemical parameterizations"?

Response 1.5

Done. See response 1.2.

Line 92: In general, Results section should present exclusively the outcomes obtained in the present study and so should not include citations to other studies. The present "3. Results" section looks more like a "Results and Discussion" section. Please consider restructuring of the manuscript.

Response 1.6

We fully understand the referee's point of view, and acknowledge that our structure somewhat differs from that of a typical paper. However, we are reluctant to change the structure for one main reason. The way we wrote the "Results" section is meant to take the reader progressively through several aspects of our simulations, each time comparing our results with reference studies to give confidence as to the performance of the model and the robustness of the subsection's results and those of the next subsections. Separating results and comparison would somehow suspend the reader's validation of our results until the discussion section, or force a tedious back-and-forth between both sections. Additionally, it seems a stand-alone section for comparing our results to existing ones would imply having a significant amount of repeat in the text.

Therefore, we stick to the initial structure. Nevertheless, to make it clear from the start, we renamed section 3 "Results and comparison to existing estimates".

Comment 1.7

Line 107: Can you specify what is "the change in empirical constraint" responsible for the larger LASC?

Response 1.7

This is specified in the following sentence (and already was in the previous version of the manuscript).

Comment 1.8

Line 223: Please explain what are the "seven categories of LUCCC activities", in a consistent manner with those in the "2. Overview of the methodology" section (only three activities in Line 69-70).

Response 1.8

Done. We have added the following text at the beginning of section 3.5.

*These categories are essentially a subdivision of the main three LULCC activities mentioned previously in the short description of OSCAR. Category 1 corresponds to land-cover change (LCC) where forest is replaced by cropland. Category 2 is LCC where forest is replaced by anything else (but forest). Category 3 is the opposite of 1 and 2: LCC where any type of land but forest is replaced by forest. Category 4 is LCC where non-forested natural land is replaced by any anthropogenic land. Category 5 is the opposite of 4. Category 6 is any LCC happening among anthropogenic land (e.g. pasture to cropland). Category 7 is the sum of wood harvest and LCC happening from any type of natural land to the same type of natural land (e.g. forest to forest). Note that because of the model's structure, the effects of shifting cultivation are included in their corresponding LCC categories.*

---

## Author Comment (AC2) · 26 May 2020

Comment 2.1

CO2 emissions from land use and land cover change (LULCC) are uncertain as they are not directly observable. In the annually updated Global Carbon Budget, two methods are used to estimate the emissions: a book-keeping approach based on LULCC data and empirical response functions, and results from process-based dynamic global vegetation models. T. Gasser and co-authors present results from a model (OSCAR) which combines book-keeping approaches with process-based modelling. There are several benefits to this approach, including an additional, constrained estimate of annual CO2 emissions from the land and a method for evaluating sources of uncertainty.

This manuscript describes some of these results. Overall I think this is a very strong manuscript, it is well written and forms an important contribution to the Global Carbon Project. I appreciated having an explanation of the model in the Appendix (although I have noted a few questions about this below). My primary concern is about the choice of constraint and sensitivity of model results to the choice, which I explain below. There a couple of other 'substantial' comments, and a few minor comments.

Response 2.1

We thank the referee for the detailed comments and for her support.

Comment 2.2

The constraint:

First, what is the constraint used in reducing the ensemble of simulations (as in, what is the value)? I see it is given at Line 112, but I think this is an important piece of information to include in the Methods and in Section A.5.

Response 2.2

This value is now given in section 2 (overview of the methodology) and in section A5.

Comment 2.3

The choice of constraint seems very important. It's not clear how robust the results are to the choice of constraint (which is important given how uncertain the constraint is). Although this is not discussed in the manuscript, some information about sensitivity of results to the constraint is provided within the figures and tables. From examining Table 2 – in general there is overlap between the annual and cumulative emissions when comparing the two methods of constraining OSCAR. The biggest differences are for cumulative land carbon sink (which I think makes sense given that the original constraint mostly impacted the land carbon sink in Figure 1c, indicating this is a large source of potential uncertainty), and for LASC (which has a strong dependence on the

land carbon sink, so that makes sense as well). I think the manuscript should include discussion of this sensitivity; and justification for choosing the constraint in this study.

Response 2.3

Thank you for this comment. We added the following paragraph in the discussion section to discuss the sensitivity and explain our choice. We note that our choice of constraint was also motivated by simplicity, as we see this part of our work more like a proof of concept that can be refined at a later stage, within the whole GCB process.

*Table 2 shows that the choice of constraint does not drastically impact our results, as there is a large overlap between the estimates obtained with both old and new constraints. More precisely, LULCC emissions do not show a large impact, whereas the land sink and the LASC do. This is however somewhat artificial, as both our constraints are aimed at constraining the processes that dictate the land sink (such as the fertilization effect), which is visible in Figure 1c where the unconstrained distribution of the land sink exhibits a large spread that is reduced after constraining. Other (or additional) constraints focused on LULCC emissions, such as constraints on carbon densities, could be envisioned – although we deemed it unnecessary for this study. Because the constraining is done after the simulations are actually run, it is indeed possible to decide ex-post on the best constraint (or combination thereof) depending on one's ultimate goal. Our choice of constraint was driven by our will to make our estimates of LULCC emissions compatible with the overall GCB2019, our scientific conviction that it is preferable to use physical (i.e. observable) variables as constraints, and our own expert judgement as to which parts of the GCB are the most reliable. Our choice can be debated, however, and we invite the community to download our raw estimates and apply their own constraints if they so wish (see data availability). Ultimately, a Bayesian synthesis framework could be used at the GCB scale (Li et al., 2016) to avoid having to make such an arbitrary choice.*

Comment 2.4

A related question I have (although maybe not for this paper): The uncertainty analysis revealed that biogeochemical parameters contribute to large proportion of the model uncertainty. This is attributed to the carbon densities (Lines 175-176), so it makes me think an additional constraint could be carbon stored in soils and vegetation. Have the authors considered using present-day carbon stocks as an additional constraint?

Response 2.4

It is entirely possible to add constraints to this setup. As written in the added text of response 2.3, It is even fairly easy to do so, because the constraining happens ex-post (i.e. after the simulations are done).

Two very much interlinked points to consider, however. First, using carbon stocks or carbon densities as constraints likely implies to do it at the regional scale, which means adding at least 10 constraints. Second, the more constraints are used, the more "dissolved" their effect, which can lead to relax all the constraints in the end. This explains why we were reluctant to do it in this study.

Ultimately, we decided to use our unique constraint as an example of the setup, leaving the question of which constraint or combination thereof is best open for a discussion within the GCP or possibly a follow-up study. Since we will provide our raw outputs, the referee or anyone can actually look into it. (Another constraint, along carbon stocks, could be NPP.)

Comment 2.5

Other comments:

Lines 435-438: The pre-industrial steady state carbon pools are determined using average climate variables from 1901-1930. I'm not sure how much IAV occurred during this time period, but using average climate, rather than looping through years (as in GCB), will neglect the impact of IAV on the pools. Is it possible this has an impact on the steady state pool values, and would cause spurious transient responses when

switching to the historical transient simulations?

Response 2.5

Thank you for this comment.

First, there may be a slight misunderstanding: the preindustrial steady-state is immediately obtained after calibration on the DGVMs (on the S0 simulation of TRENDY), i.e. we do not spin-up our model. This is now explicit in the main text, following comment 1.4. Therefore, our model simulates a deviation to that preindustrial steady-state, as discussed in (Gasser et al. 2017).

It is however true that in our transient simulations, the climate variables before 1901 (i.e. before reconstructions are available) are assumed constant and equal to the average over 1901-1920 (and not 1930; this was a mistake in our text – it is now corrected). We believe the referee's concerns are about this second aspect, as DGVMs actually recycle the first 20 years of the climate data, for this period over which we do not have data.

We made a quick test, using 2000 unweighted Monte Carlo elements, and produced the attached figure. The first line of panels shows the simulations results when using our 'flat' climate assumption or the recycled climate. The second line of panels shows the absolute difference (i.e. recycled minus flat simulations). One can see that the impact on the land sink is as expected: a cycle with >1 GtC/yr of amplitude appears before 1901. However, this virtually does not affect the LULCC emissions (which are the focus of our paper). The cumulative net change in land carbon stock is also extremely limited: it is about 1 GtC in 1901 – which is very low compared to the 40 GtC already emitted at that point – and it slowly decays to be nearly nothing after 1950. We note that this behavior is likely specific to OSCAR: it is not a process-based model, it has a yearly time-step, its sensitivities were calibrated to reproduce the effect of climate change (and not climate!), and it has a fairly linear formulation.

Therefore, we believe this confirms that our experimental setup is sound, if not exactly in line with the GCB's. And thus we will not redo our simulations. We also note that in future simulations (e.g. for future GCBs), this could easily be changed and aligned with the TRENDY exercise. Nevertheless, we added to the relevant appendix mention of this test and the fact that it does not impact our results.

Comment 2.6

Lines 284-286: Annual emissions would be less without changes in environmental conditions (according to the model): in terms of what the model is simulating, why is this? (e.g. is it increasing fire emissions? More land carbon without the changes resulting in higher emissions? Etc)

Response 2.6

This is mainly caused by a lower carbon density that is itself caused by the absence of fertilisation effect (as atmospheric $CO_2$ remains constant to preindustrial levels in this extra simulation). This was added to the corresponding section of the main text.

Comment 2.7

Minor comments OSCAR calculates carbon stocks and fluxes for average biomes within 5 regions –what are the biomes? I think these should be stated up front in Section 2 and in A.1.Also state number of regions.

Response 2.7

Done in both places.

Comment 2.8

Line 181: Typo; I think you mean Table 3.

Response 2.8

Corrected throughout section 3.3.

Comment 2.9

Lines 358-359: Could you clarify what "definition 3" and "definition B" are?

Response 2.9

Those definitions are introduced in the respective referenced papers: (Gasser and Ciais, 2013) and (Pongratz et al., 2014). This sentence was just to make it clear which definition the structure of OSCAR corresponds to, for anyone comparing our study and those two papers. It feels unnecessary, however, to detail the definitions and add another layer of complexity, especially as the definitions are somewhat dependent on other definitions introduced in their respective papers. . .

This sentence was slightly rephrased to make it clearer that the definitions are introduced in the cited papers:

*This bookkeeping approach corresponds to the "definition 3" introduced by (Gasser and Ciais, 2013) and to the "definition B" of (Pongratz et al., 2014).*

Alternatively, that sentence could be deleted entirely.

Comment 2.10

Line 383: Recommend replacing 'a fortiori', I had to look it up

Response 2.10

Simply deleted.

Comment 2.11

Line 439: The word "past" here is a little vague; in my mind it can mean either 'beyond1950' or 'further in the past than 1950'; perhaps just change to 'after' (I think that's what is meant here).

Response 2.11

Agreed.

Comment 2.12

Lines 435-455: Perhaps reference Figure 1 and related discussion here to note that there is further discussion of the effect of the constraint in the main text.

Response 2.12

Done.

Comment 2.13

Lines 451-453: It would be helpful if you state here the number of model parameters, and refer to section A.7 where they are introduced.

Response 2.13

Done.

Comment 2.14

Section A.7: Which CMIP5 experiments were used for calibrating the sensitivity functions?

Response 2.14

They are calibrated on 1pctCO2 experiments and their variants (esmFixClim1 and esmFdbk1). Information added. Note that the details of the calibration are given in Gasser et al. (2017).

Comment 2.15

Eq 19: How is tau_shift determined? Also it's not clear how Eq. 13 is used to derive Eq. 19. Is this a typo?

Response 2.15

The value of tau_shift is 15 years, taken as the average turnover time of shifting culti-vation according to Hurtt et al. (2006). We acknowledge this is an old value that could be reevaluated.

It was indeed a typo: Eq. 19 is derived from Eq. 3, i.e. the differential equation describing the biomass growth (that is exponential in OSCAR). We added these two pieces of information to the text.

Comment 2.16

I can't see where Fslash comes into the overall equations, other than the initialization step in Eq. 31.

Response 2.16

The referee is right: the slash only impacts initialization, since it is then considered to be part of the litter or soil pool. We do not treat it separately. We added the following sentence:

*The slash is not accounted for separately in OSCAR. Therefore, slash fluxes only appear at the initialization step, since this carbon is added to the litter and soil pools, and then follow the biogeochemistry of these pools.*

Comment 2.17

Sorry if I have missed it, but I also don't see an explanation of how the LASC is calcu-lated. I assume it's related to Eq. 42 but without transient LULCC?

Response 2.17

It is indeed missing! We added the following short paragraph at the end of section A7:

*In OSCAR, the LASC ($F_{\mathrm{LASC}}$) is naturally a subcomponent of the land carbon sink. It is deduced from Eq. (42) by difference to a case without transient land-cover change (i.e. with fixed preindustrial land-cover, noted $A_{\mathrm{land},0}$):*

$$F_{\mathsf{LASC}} = \sum_{i,b}(npp^{i,b} - e^{i,b}_{fire} - e^{i,b}_{harv} - e^{i,b}_{graz} - rh^{i,b}_1 - rh^{i,b}_2)(A^{i,b}_{land} - A^{i,b}_{land,0})$$

Comment 2.18

Table A2: What are the units (Mha?)?

Response 2.18

Yes. Units added.

Comment 2.19

Figure A1: Is the brown line in panel (e) mislabelled?

Response 2.19

Yes. Corrected.

[Figure]

**Fig. 1.** Figure for response 1.5

---

## Author Comment (AC3) · 26 May 2020

[revised manuscript text omitted]
 | $1.86 \pm 0.57$ | $2.20 \pm 0.48$ | $1.47 \pm 0.37$ | $213 \pm 93$ | $285 \pm 62$ | $131 \pm 38$ |
| Other deforestation | $0.55 \pm 0.26$ | $0.41 \pm 0.14$ | $0.70 \pm 0.27$ | $77 \pm 27$ | $88 \pm 26$ | $64 \pm 24$ |
| Reforestation and afforestation | $-1.36 \pm 0.49$ | $-1.70 \pm 0.38$ | $-0.96 \pm 0.22$ | $-144 \pm 99$ | $-230 \pm 48$ | $-45 \pm 11$ |
| Other natural land appropriation | $0.42 \pm 0.31$ | $0.63 \pm 0.26$ | $0.17 \pm 0.08$ | $60 \pm 33$ | $82 \pm 30$ | $35 \pm 12$ |
| Other natural land (re)establishment | $-0.21 \pm 0.18$ | $-0.33 \pm 0.15$ | $-0.07 \pm 0.08$ | $-20 \pm 18$ | $-34 \pm 11$ | $-3 \pm 2$ |
| Conversions among anthrop. biomes | $0.02 \pm 0.02$ | $0.03 \pm 0.02$ | $0.01 \pm 0.01$ | $1 \pm 1$ | $1 \pm 1$ | ~0 |
| Wood harvest | $0.09 \pm 0.04$ | $0.10 \pm 0.04$ | $0.07 \pm 0.03$ | $19 \pm 6$ | $21 \pm 5$ | $16 \pm 6$ |
| **Loss of additional sink capacity (LASC)** | | | | | | |
| Deforestation for cropland | $0.29 \pm 0.16$ | $0.31 \pm 0.17$ | $0.27 \pm 0.15$ | $14 \pm 6$ | $16 \pm 7$ | $12 \pm 5$ |
| Other deforestation | $0.24 \pm 0.15$ | $0.30 \pm 0.15$ | $0.17 \pm 0.10$ | $12 \pm 6$ | $15 \pm 6$ | $7 \pm 3$ |
| Reforestation and afforestation | $-0.15 \pm 0.10$ | $-0.21 \pm 0.10$ | $-0.09 \pm 0.04$ | $-7 \pm 5$ | $-10 \pm 4$ | $-3 \pm 1$ |
| Other natural land appropriation | $0.39 \pm 0.53$ | $0.57 \pm 0.63$ | $0.19 \pm 0.24$ | $16 \pm 21$ | $24 \pm 25$ | $8 \pm 9$ |
| Other natural land (re)establishment | $-0.09 \pm 0.12$ | $-0.14 \pm 0.14$ | $-0.03 \pm 0.06$ | $-3 \pm 4$ | $-5 \pm 5$ | $-1 \pm 1$ |
| Conversions among anthrop. biomes | $0.00 \pm 0.01$ | $0.00 \pm 0.01$ | ~0.00 | ~0 | ~0 | ~0 |
| Wood harvest | 0.00 | 0.00 | 0.00 | 0 | 0 | 0 |

[revised manuscript text omitted]

---

## Referee Report (RR1)

I've reviewed the changes made by the authors and their responses. I appreciate the additional information they've included in the discussion on the constraint, and with regards to their methods for obtaining the pre-industrial steady state. In their response, there was one remaining question about wording of text at the end of Appendix A1 (they labelled it as Comment 2.9) – I am happy with the change they made. So I have no further comments on the paper, and recommend it is accepted as is.